Review Article

EMBO
reports

# Epigenetic inheritance and gene expression regulation in early Drosophila embryos

Filippo Ciabrelli [iD][1], Nazerke Atinbayeva [iD][1], Attilio Pane [iD][2] & Nicola Iovino [iD][1]✉

## Abstract

Precise spatiotemporal regulation of gene expression is of paramount importance for eukaryotic development. The maternal-to-zygotic transition (MZT) during early embryogenesis in *Drosophila* involves the gradual replacement of maternally contributed mRNAs and proteins by zygotic gene products. The zygotic genome is transcriptionally activated during the first 3 hours of development, in a process known as "zygotic genome activation" (ZGA), by the orchestrated activities of a few pioneer factors. Their decisive role during ZGA has been characterized in detail, whereas the contribution of chromatin factors to this process has been historically overlooked. In this review, we aim to summarize the current knowledge of how chromatin regulation impacts the first stages of *Drosophila* embryonic development. In particular, we will address the following questions: how chromatin factors affect ZGA and transcriptional silencing, and how genome architecture promotes the integration of these processes early during development. Remarkably, certain chromatin marks can be intergenerationally inherited, and their presence in the early embryo becomes critical for the regulation of gene expression at later stages. Finally, we speculate on the possible roles of these chromatin marks as carriers of epialleles during transgenerational epigenetic inheritance (TEI).

**Keywords** H3K27me3; H3K9me3; Epigenetic Inheritance; Chromatin Establishment After Fertilization
**Subject Categories** Chromatin, Transcription & Genomics; Development

## Introduction

In sexually reproducing species, the embryo predominantly relies on maternally deposited products for the first stages of development. After fertilization, the zygote of *Drosophila* undergoes a series of rapid nuclear divisions within a common cytoplasm, resulting in distinct cell nuclei known as syncytial nuclei (Bownes' stages 1 and 2) (Figs. 1–3 top, for correspondence between Bownes' stage and nuclear cycles) (Campos-Ortega and Hartenstein, 1985). At nuclear cycle 8, three nuclei migrate posteriorly to form the pole cells, with the remaining somatic nuclei still lingering in the interior of the fly embryo. Up to this point, very few genomic loci show transcriptional activity (Lécuyer et al, 2007; Pérez-Mojica et al, 2023).

At nuclear cycle 9, somatic nuclei start to migrate toward the cortical region of the embryo (Bownes' stage 3). From nuclear cycle 10 onwards, the duration of the cell cycle increases and transcriptional onset after mitosis is DNA-replication dependent (Cho et al, 2022). At nuclear cycle 9, a few hundred of the zygotically expressed genes display substantial transcriptional activity, setting the stage for the minor wave of zygotic genome activation (ZGA) (De Renzis et al, 2007; Harrison and Eisen, 2015; Kwasnieski et al, 2019; Lott et al, 2011; Pérez-Mojica et al, 2023; Vastenhouw et al, 2019). These early genes are mostly driven by the activity of the pioneer transcription factor Zelda (Zld) (Harrison et al, 2011; Liang et al, 2008; Pérez-Mojica et al, 2023). Because of the quick pace of nuclear cycles, consisting of alternating S and M phases only, early genes need to be particularly short, and are often intron-less, in order to be promptly transcribed (Heyn et al, 2014; Kwasnieski et al, 2019).

At nuclear cycle 14, more than 6000 zygotic genes are de novo transcribed during the second and major wave of ZGA (De Renzis et al, 2007; Ibarra-Morales et al, 2021; Kwasnieski et al, 2019; Lott et al, 2011; Saunders et al, 2013). The expression of these genes is mediated by the activity of Zld in concert with other pioneer factors such as GAGA Factor (GAF) (Gaskill et al, 2021; Moshe and Kaplan, 2017) Clamp (Colonnetta et al, 2021; Duan et al, 2021) and Odd-paired (Opa) (Colonnetta et al, 2021; Duan et al, 2021; Koromila et al, 2020; Soluri et al, 2020). These pioneer factors act synergistically to expose DNA-binding sequences in gene regulatory elements, thus favoring the binding of transcription factors to their target genes (Iwafuchi-Doi and Zaret, 2014, 2016). We recently showed that the deposition of the histone variant H2Av (the *Drosophila* ortholog of mammalian H2A.Z and H2A.X) by the histone chaperone Domino, is required for the activation of more than 4000 Zld-independent genes at nuclear cycle 14 (Ibarra-Morales et al, 2021).

At this point, the cellularization of about 6000 cortical nuclei occurs, transforming the fly embryo from a syncytial to a cellular blastoderm architecture. The introduction of the G2 phase in the cell cycle also occurs at this stage, lengthening the cell cycle to 1 h (Bownes' stages 5, 6, and 7). Mitosis happens during the imminent 20-minute-long gastrulation process (Bownes' stages 6 and 7). The combination of this stepwise ZGA with the concomitant

[1]Max Planck Institute of Immunobiology and Epigenetics, 79108 Freiburg im Breisgau, Germany. [2]Institute of Biomedical Sciences/UFRJ, 21941902 Rio de Janeiro, Brazil.
✉E-mail: iovino@ie-freiburg.mpg.de

## Bownes stages

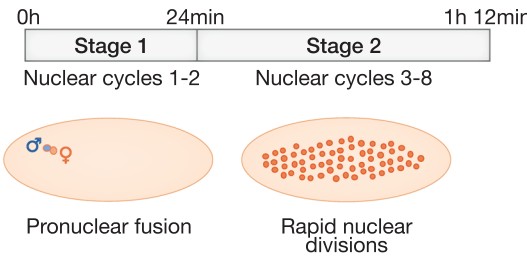

| 0h | 24min | 1h 12min |
|---|---|---|
| Stage 1 | Stage 2 | |
| Nuclear cycles 1–2 | Nuclear cycles 3–8 | |

Pronuclear fusion          Rapid nuclear divisions

## Chromatin marks/proteins

| H4K16ac | Intergenerationally transmitted | H3K4me1 | Not detected |
|---|---|---|---|
| H3K27ac | Detected towards the end of stage 2 | H3.3 | Paternal protamine substitution |
| H3K9ac | Not detected | H2Av | Not detected (only at apposition) |
| H3K27me3 | Intergenerationally inherited | BigH1 | Widespread presence |
| H3K9me3 | Intergenerationally inherited | HP1a | Detected on heterochromatin |

## Chromatin interactions

© EMBO

Unstructured chromosomes          Long-range chromatin contacts

## Nuclear organization

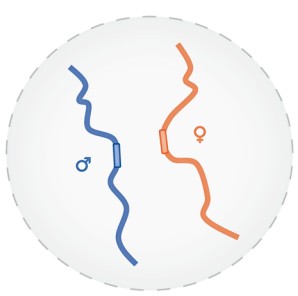

Unpaired homologous chromosomes

**Figure 1. Chromatin marks, chromatin interactions, and nuclear organization during fertilization and early divisions.**

Top left, schematic view of *Drosophila* embryos from pronuclear fusion (stage 1) to rapid nuclear divisions (stage 2). Time post fertilization, Bownes' stage and nuclear cycle are defined on the top bar. Top right, description of relevant histone marks and chromatin protein distribution during early stages of embryogenesis. Bottom left, long-range chromatin interactions during early stages of embryogenesis. The triangles represent Hi-C interaction matrices. Along the linear representation of a chromosome in black, pioneer factors are depicted in green. Bottom right, nuclear organization during early stages of embryogenesis. Homologous chromosomes (in blue and red) are initially unpaired inside the nuclei (in gray).

degradation of maternal products defines the maternal-to-zygotic transition (Hamm and Harrison, 2018).

Several excellent reviews have carefully described the current knowledge on the role of pioneer factors in *Drosophila* ZGA (Hamm and Harrison, 2018; Harrison and Eisen, 2015; Iwafuchi-Doi and Zaret, 2014, 2016; Lefebvre and Lécuyer, 2018). Considerably fewer efforts have been made to specifically integrate the literature on chromatin regulation during *Drosophila* early embryonic development (Boija and Mannervik, 2015). In this review, we gathered current knowledge on how chromatin factors contribute to the proper spatiotemporal regulation of both transcriptional activation and silencing during the early stages of embryonic development. We also discuss the establishment of constitutive heterochromatin in the early embryo. For instance, although earlier work showed that heterochromatin formation coincides with ZGA (Rudolph et al, 2007; Seller et al, 2019; Shermoen et al, 2010; Yuan and O'Farrell, 2016), more recent studies indicate that this process begins much earlier during fly development (Atinbayeva et al, 2024; Zenk et al, 2021). We summarize recent insights into how the genome starts to organize its structure inside the nucleus, through the formation of long-range chromatin contacts, topologically asscociating domains (TADs), A and B compartments and homologus chromosome pairing. Finally, we discuss the role of chromatin in intergenerational and transgenerational inheritance of epigenetic modifications.

# Fertilization and early divisions

## From nuclear cycle 1 to nuclear cycle 8

### Chromatin marks inheritance and histone replacement

Fusion of the parental haploid gametes, two highly specialized and terminally differentiated cell types, gives rise to a totipotent nucleus. This zygotic nucleus, in turn, will divide into daughter nuclei that will progressively lose the totipotent state according to their localization in the developing embryo (Campos-Ortega and Hartenstein, 1985; Loppin et al, 2015).

In a *Drosophila* egg, maternal chromosomes are arrested at the metaphase of meiosis I, with meiosis resuming upon egg activation and sperm fertilization. While maternally inherited chromatin bears canonical histones, the sperm chromatin contains highly basic protamines and is very condensed (Bao and Bedford, 2016; Loppin et al, 2015; Rathke et al, 2014). During *Drosophila* spermatogenesis, histones are almost completely replaced with at least three types of protamines (i.e., protamine-A, protamine-B, and Mst77F) (Rathke et al, 2014), with the exception of a few loci that retain canonical histones (Elnfati et al, 2016). The presence of Paternal Loss protein is necessary for (H3–H4)$_2$ tetramer eviction after H2A–H2B dimer removal and consequent protamine incorporation. Lack of the

Paternal loss protein results in paternal inheritance of $(H3–H4)_2$ tetramers, which impairs the male pronucleus integrity during female meiosis (Dubruille et al, 2023).

Once the sperm enters the egg, extensive chromatin remodeling will transform it into a mature male pronucleus. This process entails the substitution of protamines with maternally provided histones in a replication-independent manner (Emelyanov et al, 2014; Loppin et al, 2001). The paternal pronucleus is gradually decondensed with the help of Dedhead, a thioredoxin enzyme that catalyzes the reduction of disulfide bonds (Tirmarche et al, 2016). Subsequently, H3.3/H4 dimers are incorporated by the histone chaperone HIRA in cooperation with the Yemanuclein protein (Aitahmed et al, 1992; Loppin et al, 2005; Orsi et al, 2013). In turn, the histone chaperone ASF1 is required to load the H3.3–H4 dimers on the HIRA complex prior to histone deposition on paternal DNA (Horard et al, 2018). These mechanisms ensure that the paternal chromatin is marked with H3.3, whereas the maternal chromatin is associated with the canonical histone H3. The histone H3.3 variant, however, is not retained in the male pronucleus but is gradually diluted during the first nuclear divisions (Bonnefoy et al, 2007). Male pronucleus decondensation also relies on the chromodomain protein and chromatin remodeler CHD1 (Konev et al, 2007; Orsi et al, 2009), whose critical role during the first nuclear divisions was confirmed by a recent RNAi-based genetic screen (Ciabrelli et al, 2023). The sperm nucleus is also the carrier of another H3 variant dubbed CID (ortholog of mammalian CENP-A), which associates with centromeric chromatin also in the embryonic paternal chromatin (Dunleavy et al, 2012; Loppin et al, 2001). Consistent with a crucial role in determining the identity of the centromeres and regulating mitotic spindle assembly, loss of CID from sperm nuclei results in the loss of paternal chromosomes early during the cleavage phase (Raychaudhuri et al, 2012). Sperm and egg, therefore, introduce different histone variants in the early embryo, which are initially retained on the chromatin of the male and female pronuclei but are replaced by canonical histone variants while the cleavage stage unfolds.

### Constitutive heterochromatin establishment

Constitutive heterochromatin can be defined as the portion of the genome that remains condensed during interphase in every nuclear type. In *Drosophila*, constitutive heterochromatin occupies roughly one-third of the genome, and is mostly concentrated at pericentromeric regions, at telomeres, on the male Y chromosome, and on chromosome 4. In addition, constitutive heterochromatin in the form of discrete islands can also be found on euchromatic portions of the chromosome arms (Eissenberg and Reuter, 2009). Studies with micrococcal nucleases revealed that this type of chromatin is characterized by low DNA accessibility and organized nucleosomal arrays (Sun et al, 2001). Constitutive heterochromatic regions are typically repeat-rich with a low density of genes, which, in turn, are usually long, intron-rich, and associated with the histone modifications H3K9me2/me3 and H4K20me3, and Heterochromatin Protein 1a (HP1a) (Riddle et al, 2011; Schotta et al, 2002; Yasuhara and Wakimoto, 2008).

The establishment of constitutive heterochromatin has been studied for decades using the fly embryo as a model and with an evolving array of cutting-edge techniques. Pioneering studies in the early 90 s relied on the famous C-banding technique to describe the distribution of constitutive heterochromatin in early embryos. This approach showed that, with the exception of the Y chromosome, constitutive heterochromatin is not a feature of pre-blastoderm nuclei, while the banding pattern typical of alternating heterochromatic and euchromatic domains is clearly visible in ZGA embryos (Vlassova et al, 1991). Later, it was shown that satellite DNA compaction is visible already at nuclear cycle 8 (Shermoen et al, 2010), even though H3K9me2/3 enrichment, a typical hallmark of heterochromatin, was not visible at this stage (Rudolph et al, 2007; Seller et al, 2019; Yuan and O'Farrell, 2016). Interestingly, H3K9me3 can be detected at the maternal pronucleus in the mouse zygote immediately after fertilization (Arney et al, 2002; Santos et al, 2005). In line with these observations, our lab recently showed that H3K9me3 also decorates the maternal pronucleus of the fly embryo (Atinbayeva et al, 2024). Furthermore, by generating triple mutant fly embryos that lack all three maternally provided H3K9 methyltransferases (i.e., G9a, Su(var)3-9 and Eggless/dSetDB1), we demonstrated that H3K9me3 is not de novo established, rather it is intergenerationally inherited from the oocyte (Atinbayeva et al, 2024) (Fig. 1, top right). We found that H3K9me3 is actively maintained on chromatin through early nuclear division cycles (Atinbayeva et al, 2024), where it predominantly associates with transposable elements. H3K9me3 is also detectable at low levels in Japanese killifish, medaka (Oryzias latipes) embryos from early cleavage stages, similar to fly embryos (Fukushima et al, 2023). By performing ChIP-seq assays with hand-sorted embryos, we recently showed that also HP1a is bound to the fly chromatin already during the first nuclear divisions (Zenk et al, 2021). The assembly of centromeric heterochromatin requires the contribution of the homeobox protein Homothorax and when this factor is not maternally provided, the CID protein is mislocalized (Salvany et al, 2009). Interestingly, centromeric transcription is not required for CID incorporation during early embryogenesis, unlike at later developmental stages (Ghosh and Lehner, 2022).

### Chromatin-mediated transcriptional silencing

Polycomb-mediated repression represents one of the main mechanisms through which chromatin silencers repress gene expression. In flies, Polycomb repressive complex 1 (PRC1) is composed of Pc (Polycomb), Ph (Polyhomeotic), Psc (Posterior Sex Combs) and Sce (Sex Combs Extra), and Polycomb repressive complex 2 (PRC2) contains the core components E(z) (Enhancer of Zeste), Su(z)12, Esc and Caf1-55 (Kassis et al, 2017). Sce is the catalytic subunit of PRC1 and can ubiquitinylate the H2AK118 residue (Wang et al, 2004), whereas E(z) is the catalytic subunit of PRC2 and is responsible for tri-methylating the H3K27 residue.

Histone replacement studies in both flies (Leatham-Jensen et al, 2019; Pengelly et al, 2013) and mammals (Sankar et al, 2022) revealed that E(z) histone methyltransferase activity is strictly required for its function. In this respect, E(z) differs from Nejire (ortholog of mammalian CBP and p300 proteins), Gcn5, Trr and other histone modifiers (Cao et al, 2002; Ciabrelli et al, 2023; Czermin et al, 2002; Müller et al, 2002; Rickels et al, 2017; Sankar et al, 2022). Similarly, the catalytic activity of Sce is not strictly required for the repression of canonical Polycomb target genes (Pengelly et al, 2015). Recent findings suggest that H2AK118ub facilitates H3K27me3 deposition but simultaneously antagonizes Polycomb-mediated repression, making its balance crucial for gene regulation (Bonnet et al, 2022).

In 2017, work from our laboratory showed that H3K27me3 in flies is intergenerationally transmitted from the maternal germline, inherited by the female pronucleus, and maintained throughout the nuclear divisions, an evolutionary conserved mechanism identified also in mammals (Inoue, 2023). While the majority of the H3K27me3 domains only appeared during the minor wave of ZGA, thirty-two H3K27me3-rich domains could be already observed at the early stages of embryonic development and appeared to propagate until ZGA (Zenk et al, 2017). Defects in H3K27me3 maternal transmission resulted in the misregulation of Hox genes and embryonic lethality, and these phenotypes could not be rescued by zygotic expression of *E(z)*. Notably, H3K27ac deposited by Nej accumulated where H3K27me3 was depleted, especially at enhancer regions. Most of these enhancers that aberrantly acquired higher levels of H3K27ac upregulated their corresponding genes at nuclear cycle 14, but not earlier (Zenk et al, 2017). We concluded that the presence of H3K27me3 in wild-type embryos is necessary to protect enhancer regions from spurious Nej binding and its coactivator function. Nevertheless, at ZGA some PREs are also bound by Nej, suggesting chromatin remodeling at this stage and most likely in a cell type-specific fashion (Hunt et al, 2022). Therefore, correct H3K27me3 deposition guarantees proper gene expression during ZGA. Histone replacement studies confirmed the centrality of the H3K27me3 mark and its instructive nature (McKay et al, 2015; Pengelly et al, 2013). Taken together, these findings suggest that chromatin-mediated transcriptional repression represents the default state during early embryonic development. Transcriptional activators, with the help of chromatin factors, likely need to overcome this silencing barrier at specific loci with the correct timing, in order to achieve precision in transcriptional regulation.

### Chromatin-mediated transcriptional activation

During the early cleavage stage of development, the genome undergoes rapid cycles of DNA replication and nuclear division without intervening gap phases. Overall, the chromatin is not yet transcriptionally competent at this stage. An exception to the rule is represented by some transposable elements (i.e., *copia, Doc, Ste12DOR*), whose mRNAs are transcribed already at stages 1 and 2 (Lécuyer et al, 2007). Toward the end of stage 2, a subset of early expressed genes such as gap genes (e.g., *giant, tailless*, and *knirps*), genes involved in sex determination (e.g., *runt, scute*, and *sisterless A*) (ten Bosch et al, 2006), and in cellularization (e.g., *bottleneck, dunk*, and *nullo*) (Ali-Murthy et al, 2013; Pérez-Mojica et al, 2023) begin to be transcribed, slightly anticipating the minor wave of ZGA. Among these genes, some (e.g., *deadpan*) are transcribed as early as nuclear cycle 6 (Pérez-Mojica et al, 2023). Accordingly, active chromatin modifications (e.g., H3K4me3, H3K36me3, H3K9ac) are virtually absent from early embryonic chromatin (Chen et al, 2013; Li et al, 2014; Samata et al, 2020).

A notable exception is H4K16ac. The deposition of this histone modification is accomplished by the evolutionarily conserved histone acetyltransferase (HAT) Male absent on the first or MOF (Feller et al, 2015). MOF resides in the MSL and NSL complexes. In flies, the MSL complex is responsible for dosage compensation (Hilfiker et al, 1997), while the NSL complex is responsible for the activation of thousands of housekeeping genes (Sheikh et al, 2019). Strikingly, H4K16ac is intergenerationally transmitted from the mother and persists throughout early embryogenesis (Samata et al, 2020). Epigenetically transmitted H4K16ac primes the activation of NSL-regulated housekeeping genes by inducing nucleosome accessibility at thousands of genes before ZGA. Consistent with this function, the expression of post-zygotic genes is impaired when H4K16ac is not maternally deposited in oocytes. Within the MSL complex, MOF is responsible for upregulating the expression of X-linked genes exclusively in male flies in order to compensate for gene dosage between XY males and XX females. In accordance, maternal MOF depletion results in a sharp decrease in chromatin accessibility (Samata et al, 2020), and 80% of maternal MOF mutants are embryonic lethal, with all the adult survivors being females. In contrast with the promoters of post-zygotic NSL-regulated genes, the promoters of Zld-dependent genes are devoid of H4K16ac (Samata et al, 2020). Histone replacement studies have confirmed the centrality of H4K16ac (Copur et al, 2018). In conclusion, during the first nuclear divisions the genome is not conducive for pervasive expression, despite the accumulation of active epigenetic marks at some loci.

### Chromatin architecture

The three-dimensional organization of the eukaryotic genome shapes the genetic material in the constrained nuclear space and helps to coordinate fundamental processes, such as transcription and DNA replication (Bonev and Cavalli, 2016). In *Drosophila*, as in other eukaryotes, chromosomes occupy subnuclear regions defined as chromosome territories (Cremer and Cremer, 2010).

Within the territories, transcriptionally active regions show preferential long-range interactions with other active regions. Similarly, inactive regions tend to cluster with other inactive domains. The subnuclear distribution of active and inactive regions establishes the so-called "A" and "B" compartments, respectively (Lieberman-Aiden et al, 2009). Improvements in Hi-C resolution led to the additonal division of A and B compartments into subcompartments (Rao et al, 2014; Spracklin et al, 2023). The B compartment was sub-divided into three inactive subcompartments characterized by the following histone modifications and protein factors in human cells: 1) H3K9me3 and HP1α and β; 2) H3K9me2 and H2A.Z and 3) H3K27me3 and PRC2 (Spracklin et al, 2023).

At the core of the 3D chromatin organization, we find the topologically associating domains (TADs), linear genomic units that strongly prefer to engage in internal interactions (Szabo et al, 2018). TADs are often composed of developmentally coregulated genes and usually reside in a coherent chromatin environment (Ciabrelli and Cavalli, 2015; Szabo et al, 2018). In *Drosophila*, TADs were first identified in late embryos (Sexton et al, 2012). They usually span tens of kilobases (Ramírez et al, 2018; Wang et al, 2018), with active TADs being on average smaller in size than inactive ones (i.e., Polycomb-repressed, heterochromatic, or Lamina-associated TADs).

During the first stages of *Drosophila* embryogenesis, the chromosomes are rather unstructured, and neither stable long-range interactions nor TADs are formed yet. Before nuclear cycle 9, chromosomes show near-random folding and no signs of compartmentalization (Hug et al, 2017; Ogiyama et al, 2018) (Fig. 1, bottom). In addition, during the first nuclear cycles somatic chromosome pairing is still absent (Fig. 1, bottom). *Drosophila melanogaster*, like other dipteran insects, displays homolog pairing of somatic chromosomes, which consists of the spatial association of homologous chromosomes inside the nuclear space (McKee,

2004). Studies in flies demonstrated that homolog pairing is very important as it affects the regulation of gene expression through molecular mechanisms like trans-silencing (Henikoff and Dreesen, 1989) and transvection (Duncan, 2002; Gubb et al, 1990). Homolog pairing is pervasive in larval and adult tissues (95% of pairing in larval wing discs), but it is weaker during embryogenesis (Fung et al, 1998). During the first nuclear divisions, pairing is mostly undetectable and it starts to manifest during the minor wave of ZGA (Fung et al, 1998). In conclusion, at these early stages, chromatin architecture is still not strictly structured, chromosome arms are not paired and TADs are not yet defined.

## Minor ZGA

### From nuclear cycle 9 to nuclear cycle 13

#### Constitutive heterochromatin establishment

H3K9me2/3 is a hallmark of constitutive heterochromatin. Previous studies showed that the H3K9me2/3 signal in *D. melanogaster* can be detected as early as the interphase of nuclear cycles 12–13 (Seller et al, 2019; Yuan and O'Farrell, 2016). However, our recent work revealed the H3K9me2/3 mark from fertilization onwards (Atinbayeva et al, 2024). H3K9me2/3 coverage and peak number increase dramatically during minor ZGA compared to early nuclear cycles1-8 (Atinbayeva et al, 2024). Similar to *D. melanogaster*, a recent study conducted in *Drosophila miranda* showed that the number of H3K9me3 peaks dramatically increases from nearly 1000 at stage 3 (nuclear cycle 9) to more than 50,000 at stage 4. H3K9me3 peaks can be detected already at nuclear cycle 9 on TEs (Wei et al, 2021). Interestingly, however, H3K9me3 deposition at earlier stages was not tested in this study, leaving the doors open for the possibility that this mark might be deposited earlier also in *D. miranda* as we reported for *D. melanogaster* (Atinbayeva et al, 2024; Wei et al, 2021).

Few HP1a loci are detectable at interphases of cycles 11–13, but HP1a enrichment could not be detected at every heterochromatic locus analyzed (Yuan and O'Farrell, 2016). HP1a binding to chromatin at these stages was confirmed by ChIP-seq experiments in hand-sorted embryos (Zenk et al, 2021). HP1a was found to associate both with pericentromeric heterochromatin and heterochromatic islands interspersed in euchromatin at minor ZGA. Its binding levels are comparable between minor ZGA and early nuclear divisions. Moreover, HP1a-bound regions do not seem to be broader during minor ZGA compared to earlier stages (Zenk et al, 2021). HP1a levels on chromatin dramatically increase from cycle 12 to cycle 14 (Seller et al, 2019), and HP1a can form phase-separated liquid droplets detectable from the interphase of cycle 11 onwards (Strom et al, 2017).

#### Chromatin-mediated transcriptional silencing

Maternally inherited H3K27me3 is detected on many domains during the minor wave of ZGA (Zenk et al, 2017). At this stage, the presence of this histone modification is necessary for Polycomb-mediated repression of certain gap genes such as *giant* (Pelegri and Lehmann, 1994), *knirps* (Pelegri and Lehmann, 1994), *tailless* (Liaw, 2022), and *Krüppel* (Mckeon et al, 1994). Pair-rule and segment-polarity genes, whose expression precedes and determines Hox gene expression patterns, are also regulated by Polycomb

group proteins (Mckeon et al, 1994). PRC1 and PRC2 are both necessary for Polycomb-mediated repression already during these early stages. In line with our findings (Zenk et al, 2017), a recent study showed that homeotic genes are enriched with H3K27me3 way before the major ZGA, confirming the conclusion that Polycomb-mediated repression is the default state on these loci at this stage (Ghotbi et al, 2021). The repressed chromatin state established by Polycomb-mediated deposition of H3K27me3 is then reprogrammed through the precise spatiotemporal activation of gene expression during embryonic development, occuring in a segment-specific fashion.

Genome-wide transcriptional repression is also achieved by the deposition of BigH1, the only H1 histone variant so far characterized in *Drosophila*. This histone variant occupies the whole genome from the early stages till nuclear cycle 14 (Pérez-Montero et al, 2013). Histone H1 is an evolutionary conserved chromatin protein, which binds both to intranucleosomal DNA and to linker DNA and contributes to the formation of higher-order chromatin structures (Prendergast and Reinberg, 2021; Robinson and Rhodes, 2006; Zhou et al, 2013). H1 displays a tripartite structure consisting of a central globular domain, which interacts with DNA, and of the intrinsically disordered Lysine-rich amino terminal (NTD) and carboxy-terminal (CTD) domains (Allan et al, 1980). BigH1 shares the same tripartite structure as H1, with the most substantial difference lying in the acidic nature of its NTD.

At fertilization, *Drosophila* embryos are loaded with BigH1, whose spreading on chromatin prevents premature transcription. The replacement of BigH1 with canonical H1 in nuclear cycle 14 embryos paves the way to the major wave of ZGA (Pérez-Montero et al, 2013) (Fig. 2, top). *BigH1* null mutants die at cellularization, displaying increased transcriptional activity and acceleration of ZGA, together with a broad range of developmental defects such as asynchronous nuclear divisions, mitotic defects, and nuclear mislocalization (Pérez-Montero et al, 2013). These phenotypes are accompanied by high levels of DNA damage. Ectopic expression of BigH1 in *Drosophila* S2 cells revealed that BigH1 incorporation competes with H1 binding, reduces nucleosome repeat length, and interferes with RNA PolII binding (Climent-Cantó et al, 2020). These functions are strictly dependent on the acidic ED domain within the BigH1 NTD. Interestingly, BigH1 replacement with H1 results in increased nuclear volume. In addition, nucleosomes formed in the presence of BigH1 are more stable than those assembled with canonical H1 (Henn et al, 2020). Following BigH1 substitution, there is, on average, one H1 molecule per nucleosome during ZGA (Bonnet et al, 2019). In conclusion, the presence of BigH1 provides another layer of default chromatin silencing that will be overcome by its replacement with canonical H1 during ZGA. In contrast with H3K27me3, however, BigH1 is distributed on the whole genome. These two mechanisms differ in their role in the selective silencing of specific loci (i.e., H3K27me3), which can be overcome by specific transactivators, versus a broader repression (i.e., BigH1) that is released with a precise temporal control.

#### Chromatin-mediated transcriptional activation

The minor wave of ZGA ensues from the expression of a few hundred zygotic genes (Harrison and Eisen, 2015; Kwasnieski et al, 2019; Pérez-Mojica et al, 2023; Vastenhouw et al, 2019). The

## Bownes stages

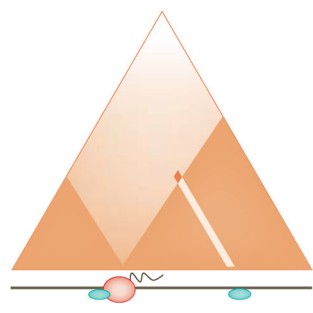

| | | | | |
|---|---|---|---|---|
| H4K16ac | Increasing on male X chromosome | H3K4me1 | Bearly detected |
| H3K27ac | Present and increasing | H3.3 | Bearly detected |
| H3K9ac | Detected towards the end of stage 4 | H2Av | Detected towards the end of stage 4 |
| H3K27me3 | Present and increasing | BigH1 | Present and decreasing |
| H3K9me3 | Present and increasing | HP1a | Present and increasing |

## Chromatin marks/proteins

## Chromatin interactions

Establishment of TAD borders

## Nuclear organization

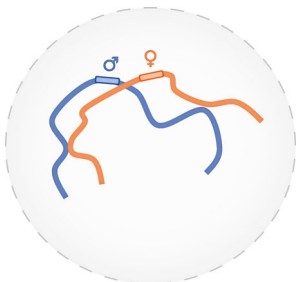

Low levels of homologous pairing

© EMBO

**Figure 2. Chromatin marks, chromatin interactions, and nuclear organization during the minor wave of ZGA.**

Top left, schematic view of *Drosophila* embryos from nuclear peripheral migration (stage 3) to nuclear cycle lenghtening (stage 4). Time post fertilization, Bownes' stage and nuclear cycle are defined on the top bar. Top right, description of relevant histone marks and chromatin protein distribution during stage 3 and 4 of embryogenesis. Bottom left, long-range chromatin interactions during stage 3 and stage 4 of embryogenesis. The triangles represent Hi-C interaction matrices. Along the linear representation of a chromosome in black, pioneer factors are depicted in green and RNA PolII in white within a red circle. Bottom right, nuclear organization during the minor wave of ZGA. Homologous chromosomes (in blue and red) slowly begin to pair inside the nuclei (in gray).

expression of these zygotic genes mostly relies on the activity of the pioneer factor Zld (Harrison et al, 2011; Liang et al, 2008; Pérez-Mojica et al, 2023), while the role of chromatin modifiers in their regulation has not been fully uncovered. ChIP-based studies showed that H3K18ac, H3K27ac, and H4K8ac are already present in early embryos, and their distribution gradually increases at later stages (Li et al, 2014) (Figs. 1 and 2, top). These histone modifications are deposited by Nej, a HAT ortholog of mammalian CBP and p300 proteins (Feller et al, 2015). Our recent work confirmed the Nej-dependent presence of H3K18ac and H3K27ac during the first wave of ZGA and, albeit to much lower levels, also at earlier nuclear cycles (Ciabrelli et al, 2023). The accumulation of these histone modifications may follow the transcription of very early genes, or it may indicate Nej binding at certain loci before RNA PolII activity. In line with our results, Nej was recently found to promote RNA PolII clustering downstream of Zld at cycle 12 (Cho and O'Farrell, 2023).

The other major *Drosophila* HAT is Gcn5, the catalytic subunit of the SAGA, ATAC, ADA and CHAT complexes (Helmlinger et al, 2021; Torres-Zelada and Weake, 2021). Gcn5 is responsible for the acetylation of H3K9 and H3K14 residues in flies (Feller et al, 2015), although the latter acetylation can also be introduced by Chameau in specific developmental contexts (Regadas et al, 2021). During

embryogenesis, the Gcn5-containing CHAT complex is responsible for global H3K14ac (Torres-Zelada et al, 2022). In contrast to Nej-dependent acetylation of histones, very few H3K9ac peaks could be called at nuclear cycle 12 (Li et al, 2014). Recently, we have shown that low levels of H3K9ac can already be detected during the first wave of ZGA (Ciabrelli et al, 2023). However, the Gcn5-dependent histone mark is detected on chromatin later than the Nej-dependent ones (e.g., H3K27ac), in line with the specific role of Nej in activating early Zld-dependent genes (Ciabrelli et al, 2023).

While Nej- and Gcn5-dependent acetylation marks start to appear during the minor wave of ZGA, the maternally maintained H4K16ac is already present on chromatin from early nuclear cycles (Samata et al, 2020) (Figs. 1 and 2, top). From the zygote stage and throughout embryogenesis, this histone modification is associated with constitutively transcribed autosomal genes, where it ensures chromatin accessibility at promoter regions before their transcriptional activation (Samata et al, 2020). Interestingly, promoters of early genes transcribed during minor ZGA are completely devoid of H4K16ac (Samata et al, 2020), indicating that MOF-dependent and Zld-dependent types of regulation are independent of each other.

H4K16ac is also deposited by the MSL (male-specific lethal) complex on the male X chromosome in order to achieve dosage compensation through 2-fold transcriptional upregulation. The

MSL complex consists of at least five proteins (MSL1, MSL2, MSL3, MLE and MOF) and either the *roX1* or the *roX2* long noncoding RNAs (Shevelyov et al, 2022). The MSL complex is initially recruited at roughly 20 sites along the male X chromosome through the binding to GA-rich MSL2 recognition elements (MREs). Later, it spreads to nearby expressed genes (Alekseyenko et al, 2008; Straub et al, 2008). A diffuse X chromosome territory coordinated by MSL2 is first detectable during the major ZGA in XY embryos. However, MOF accumulation on the male X chromosome begins only at the gastrulation stage, just after the major ZGA (Samata et al, 2020). This stepwise pattern coincides with *roX1* and *roX2* expression, starting during minor and major ZGA, respectively (Lott et al, 2011; Meller and Rattner, 2002). Although MOF can be detected on autosomes already during the minor wave of ZGA (Samata et al, 2020), higher H4K16ac levels at the male X chromosome compared with the autosomes are only detectable from the late gastrulation stage onwards (Franke et al, 1996; Rastelli et al, 1995; Rieder et al, 2019; Samata et al, 2020). In summary, even though MSL-mediated dosage compensation of the male X chromosome starts after cellularization, assembly and recruitment of the MSL complex are progressively established already during ZGA. Meanwhile, constitutively expressed genes display both the H4K16ac mark and an open-chromatin conformation at their promoters, even though they are not transcribed yet at the minor ZGA stage.

### Chromatin accessibility

The correct supply of maternal histones to the developing *Drosophila* oocyte ensures proper chromatin accessibility and gene expression. Indeed, decreased levels of maternal histones result in accelerated ZGA, cell cycle lengthening, and premature gastrulation (Chari et al, 2019). Conversely, the overexpression of histones in the female germline, like in the *abnormal oocyte* maternal effect mutant, causes embryonic lethality (Berloco et al, 2001), delayed transcription, and introduction of an extranuclear cycle before gastrulation (Chari et al, 2019). In order to study chromatin accessibility, ATAC-seq (Assay for Transposase-Accessible Chromatin using sequencing) assays (Buenrostro et al, 2013) were conducted during minor and major waves of ZGA in fly embryos (Blythe and Wieschaus, 2016). ATAC-seq performed in nuclear cycles 11 to 13 embryos revealed that approximately one-third of all the open-chromatin regions were already accessible at nuclear cycle 11 and persisted till the major wave of ZGA, whereas the remaining two-thirds were only established later (i.e., cycles 12 and 13).

Pioneer factors are responsible for the earliest detected open-chromatin regions during *Drosophila* embryogenesis (Blythe and Wieschaus, 2016; Schulz et al, 2015; Sun et al, 2015). More specifically, Zld-dependent open regions become accessible earlier than GAF-dependent ones (Blythe and Wieschaus, 2016). Similarly, the pioneer factor Clamp binds its targets and contributes to chromatin accessibility during the first wave of ZGA (Duan et al, 2021). Zld, GAF and Clamp can bind their motifs in a nucleosomal context (Duan et al, 2021; McDaniel et al, 2019; Sun et al, 2015), mediate chromatin opening (Duan et al, 2021; Gaskill et al, 2021; Schulz et al, 2015; Sun et al, 2015), and facilitate the recruitment of canonical transcription factors to their binding sites (Brennan et al, 2023; Foo et al, 2014; Kanodia et al, 2012; Reeves and Stathopoulos, 2009).

The number and relative position of Zld binding sites are predictive of its capability to dissipate nucleosomal barriers on enhancers (Foo et al, 2014; Sun et al, 2015) and its action has an effect specifically on the local chromatin environment (Sun et al, 2015). In particular, Zld binding leads to the depletion of 1 or 2 nucleosomes, whereas local Zld binding at multiple sites leads to a broader depletion of nucleosomes (Foo et al, 2014). Zld accumulation throughout ZGA is necessary for its activity (McDaniel et al, 2019), even though, its binding to chromatin is lost during mitosis (Dufourt et al, 2019a; Dufourt et al, 2019b). Different from Zld, the GAF protein is instead retained on mitotic chromosomes, where it exerts a key role as an epigenetic bookmark during early embryogenesis (Bellec et al, 2022; Bellec et al, 2018). In the future, it would be of great interest to understand what chromatin loci become accessible first and how early in development this process occurs.

### Chromatin architecture

Although TADs are not yet formed during the minor ZGA wave, networks of long-range chromatin interactions are already established at this stage and are conserved in different germ layers (Espinola et al, 2021; Ogiyama et al, 2018). Interestingly, the loci that are involved in early long-range chromatin interactions do not correspond to future TAD borders (Messina et al, 2023) (Fig. 2, bottom). Zld is required for the formation of several chromatin hubs during these stages, connecting different Zld-rich regions with each other or with target promoters (Espinola et al, 2021; Ogiyama et al, 2018). Indeed, Zld-bound regions coalesce during the minor wave of ZGA and create hubs of active genes (Ogiyama et al, 2018).

Overall, during the minor wave of ZGA, chromatin starts to become progressively organized. At nuclear cycle 12, nearly 180 regions enriched in housekeeping genes engage in the establishment of structures that resemble TAD boundaries (Hug et al, 2017; Ogiyama et al, 2018). At nuclear cycle 12, TAD borders correlate with the presence of Nej-dependent acetylation marks (i.e., H3K18ac, H3K27ac, and H4K8ac), while H3K9ac and "active" methylation marks (i.e., H3K4me3, H3K36me3) will appear on TAD boundaries only at nuclear cycle 14 (Hug et al, 2017). Nonetheless, definitive TAD structures will emerge only later, during the major wave of ZGA (Fig. 3, bottom).

During the minor wave of ZGA, homologous chromosomes start to progressively pair (Fig. 2, bottom). At nuclear cycle 13, homolog pairing at euchromatic regions reaches an average of 10%. However, there is consistent variability between different loci (Fung et al, 1998). Interestingly, some loci, such as the histone locus, seem to pair before others, displaying 61% pairing at nuclear cycle 13, while repeat-rich regions tend to become paired less extensively than non-repetitive regions at this stage (Fung et al, 1998). More recent studies used a haplotype-resolved Hi-C to study embryonic pairing and revealed that interactions between homologous chromosomes spread genome-wide at the time when pairing begins (Erceg et al, 2019). Moreover, the presence of Zld, but not of GAF binding, correlates with the establishment of homologous pairing at this stage (Erceg et al, 2019). In summary, chromatin activation progressively transitions from random folding into a partially ordered structure during the minor ZGA wave, in a process that seems to be mostly coordinated by the pioneer factor Zld.

## Bownes stages

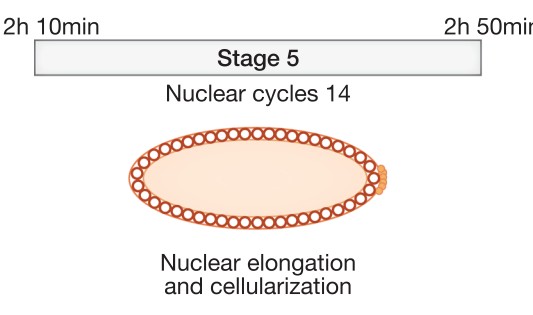

2h 10min                                    2h 50min

Stage 5

Nuclear cycles 14

Nuclear elongation
and cellularization

## Chromatin marks/proteins

| H4K16ac | Present on male X and autosomes | H3K4me1 | Widespread presence |
| --- | --- | --- | --- |
| H3K27ac | Present on every active gene | H3.3 | Present on every active gene |
| H3K9ac | Present on every active gene | H2Av | Present on thousands of active gene |
| H3K27me3 | Present and increasing | BigH1 | Not detected |
| H3K9me3 | Present on hetero-chromatin and chromosome arms repeats | HP1a | Present on hetero-chromatin and chromosome arms |

## Chromatin interactions

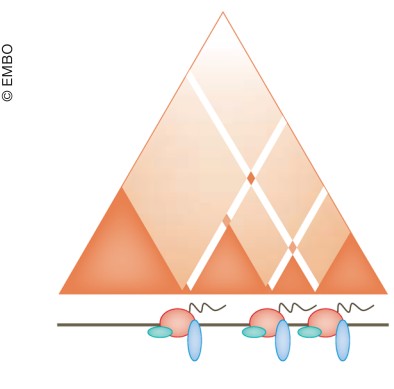

© EMBO

TAD formation

## Nuclear organization

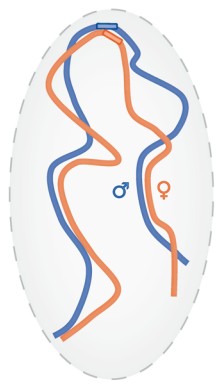

♂ ♀

• "Rabl" conformation
• High levels of homologous
  pairing

**Figure 3. Chromatin marks, chromatin interactions, and nuclear organization during the major wave of ZGA.**

Top left, schematic view of *Drosophila* embryos during nuclear elongation and cellularization (stage 5). Time post fertilization, Bownes' stage and nuclear cycle are defined on the top bar. Top right, description of relevant histone marks and chromatin protein distribution during stage 5 of embryogenesis. Bottom left, long-range chromatin interactions during stage 5 of embryogenesis. The triangles represent Hi-C interaction matrices. Along the linear representation of a chromosome in black, pioneer factors are depicted in green, insulator proteins in blue and RNA PolII in white within a red circle. Bottom right, nuclear organization during major wave of ZGA. Homologous chromosomes (in blue and red) are mostly paired inside the elongated nuclei (in gray).

## Major ZGA

### Nuclear cycle 14

#### Constitutive heterochromatin establishment

Constitutive heterochromatin is clearly visible during the major wave of ZGA. At this stage, constitutive heterochromatin has segregated from euchromatic regions within the nuclear space (Zenk et al, 2021), and the C-banding technique highlighted clear bands in ZGA embryos (Vlassova et al, 1991).

H3K9me2/3 levels sharply increase at major ZGA (Atinbayeva et al, 2024; Seller et al, 2019; Yuan and O'Farrell, 2016). At pericentromeric regions of the current genome version (dm6), H3K9me3 is fully established at major ZGA (Atinbayeva et al, 2024). H3K9me2/3 marks are critical for embryonic development as triple mutant embryos that lack H3K9me2/3 display strong defects in embryogenesis. In contrast, single mutants for each H3K9 methyltransferase do not display strong embryonic

developmental defects. Intriguingly, the phenotypes of the triple mutant embryos can be rescued by maternal, but not zygotic Eggless/dSetDB1 expression, pointing out that H3K9me2/3 presence at early stages is required for embryonic development (Atinbayeva et al, 2024). Eggless/dSetDB1 is the primary methyltransferase involved in the H3K9me3 deposition at pericentromeric regions during ZGA (Atinbayeva et al, 2024). The presence of H3K9me2/3 mark itself, but not Eggless/dSetDB1 protein per se, is important for embryonic development as catalytic inactive Eggless/dSetDB1 embryos exhibit similar embryonic developmental defects as the triple mutant embryos (Atinbayeva et al, 2024).

HP1a binding at pericentromeric regions significantly increases during major ZGA (Zenk et al, 2021). In contrast, HP1a enrichment declines but does not disappear on chromosome arms at this stage (Zenk et al, 2021). Indeed, more than 2000 HP1a peaks (one-third of the total) could be detected along chromosome arms at nuclear cycle 14 (Zenk et al, 2021). Consistent with these observations, during the transition from minor to major ZGA,

HP1a peaks progressively accumulate at pericentromeric hetero-chromatin, but they do not spread along the chromosome arms (Zenk et al, 2021). Moreover, at major ZGA, HP1a is suggested to condense in phase-separated droplets exhibiting less circularity and an increased immobile fraction as the interphase of cycle 14 proceeds (Strom et al, 2017). This phenomenon could be coupled to the progressive assembly of the chromocenter (Shermoen et al, 2010). Interestingly, HP1a binding to chromatin is not completely dependent on H3K9me2/3 given that triple mutant embryos that lack H3K9me2/3 still retain HP1a bound to chromatin (Atinbayeva et al, 2024; Yuan and O'Farrell, 2016). Instead, H3K9me2/3 deposition is required for HP1a foci/phase-separated liquid droplet formation and growth (Atinbayeva et al, 2024). Indeed, the role of HP1a during major ZGA seems to be mostly structural. HP1a mediates the clustering and condensation of constitutive hetero-chromatin at pericentromeric regions and contributes to the spatial compartmentalization of inactive pericentromeric regions. In contrast, the lack of HP1a does not affect RNA Poll II activity on zygotic genes (Zenk et al, 2021).

Interestingly, even though the volume of DAPI dense regions in H3K9me2/3-depleted *Drosophila* mutant embryos decreases, it does not completely disappear, implying that additional mechanisms might be involved in constitutive heterochromatin formation in early embryos (Atinbayeva et al, 2024). A possible candidate protein that might explain these observations is the AT-hook containing protein D1. D1 was shown to be enriched at satellite repeats, and its mRNA and protein reduction leads to suppression of Position Effect Variegation (PEV) (Aulner et al, 2002; Elgin and Reuter, 2013). Accordingly, chromocenters are affected in mutants for D1 and for another satellite-binding protein known as Prod at later developmental stages (Jagannathan et al, 2019). Recent studies have started deciphering the players involved in the establishment of constitutive heterochromatin in the early stages of *D. melanogaster* embryonic development. For instance, the H3K4me2 demethylase Lsd1 was reported to be necessary for the deposition of H3K9me2/3 (Rudolph et al, 2007). Also, the insulator protein and pioneer factor GAF exert an important role in heterochromatin formation at specific loci during early embryogenesis. This protein binds to AAGAG satellite repeats, contributing to their silencing and heterochromatin establishment (Gaskill et al, 2023). Furthermore, maternally deposited Piwi and piRNAs can target *roo* transposons, silence their expression, and deposit H3K9me3 starting at ZGA until later stages of embryonic development (Fabry et al, 2021). Further studies are, however, required to fully unveil the mechanisms by which HP1a is recruited to chromatin in early embryos as well as to dissect the role of H3K9me3-dependent HP1a foci/condensates in heterochromatin formation and embryonic development.

### Chromatin-mediated transcriptional silencing

Thousands of genes are de novo transcribed during the major wave of ZGA, while the maternally provided pool of mRNAs is actively degraded in the embryo (Hamm and Harrison, 2018). At the same time, transcriptional repressors and corepressors silence the transcription of non-ZGA genes and ZGA genes that have to be specifically turned off in some regions of the embryo. Indeed, the ectopic expression of early genes involved in axes patterning, segment specification or germ-layer formation results in major

developmental defects, which often lead to lethality (Campos-Ortega and Hartenstein, 1985).

Among the mechanisms of chromatin-based gene silencing acting at ZGA, Polycomb-mediated repression is the most prominent and the best described so far (Schuettengruber et al, 2017). Most of the current knowledge on Polycomb-mediated silencing in flies derives from studies of Hox gene regulation (Ingham, 1983; Struhl and Akam, 1985; Wedeen et al, 1986). The homeotic genes harbored in the *bithorax* and the *Antennapedia* complexes display specific expression patterns, which determine parasegment identity (Maeda and Karch, 2006). The levels of H3K27me3 correlate with the establishment of silent regions along the anterior-posterior axis. Anterior parasegments are all repressed and marked with H3K27me3, whereas the most posterior parasegments lack H3K27me3 domains on *bithorax* and on the *Antennapedia* genes (Maeda and Karch, 2006). Moving from the most anterior toward the posterior parasegments, H3K27me3 domains are progressively lost on these loci. H3K27me3 absence correlates with a transcriptional active state and consequently with parasegment identity (Bowman et al, 2014). Surprisingly, however, PRC1 binding does not mirror H3K27me3 distribution at different parasegments. Similarly, the PRC2 component Su(z)12 parallels PRC1 binding, thus rendering the H3K27me3 histone mark the only discriminant for parasegment-specific silencing (Bowman et al, 2014). How PRC2 catalytic activity is stimulated at certain loci or inhibited at others is still an open question. Recent studies indicate that Polycomb repression represents the default state in early embryos (Zenk et al, 2017). Hence, we speculate that the presence of activators or histone demethylases would probably interfere with E(z) activity during ZGA (Ghotbi et al, 2021). Additional questions still await an answer. For instance, what is PRC1 function on silenced genes? Why is its presence insufficient to repress transcription in the absence of H3K27me3? Apparently, PRC1 alone can taper the expression of some genes (Loubiere et al, 2016), but it could also stimulate gene expression in different contexts. For example, it might facilitate Spt5 association with enhancers (Pherson et al, 2017) and promote enhancer-promoter interactions (Loubiere et al, 2020). PRC1 presence is necessary, but not sufficient, for chromatin compaction at Hox genes and other targets during ZGA (Cheutin and Cavalli, 2014). Strikingly, the absence of Ph or Pc affects the chromatin compaction of Hox clusters prior to ectopic transcription of Hox genes. Therefore, open-chromatin states determined by the loss of PRC1 are the cause and not the consequence of transcriptional activity (Cheutin and Cavalli, 2018).

Besides Polycomb-mediated repression, alternative silencing mechanisms take place during *Drosophila* major ZGA. For instance, the histone deacetylase HDAC1, also known as Rpd3, is responsible for correct segmentation through cooperation with the pair-rule gene protein product Even-skipped (Mannervik and Levine, 1999) and the gap gene protein product Knirps (Struffi and Arnosti, 2005). The importance of HDAC1 during ZGA has been recently confirmed. Maternal depletion of HDAC1 misregulates hundreds of genes during major ZGA (Ciabrelli et al, 2023). HDAC1 cooperates with the mesoderm-specific factor Snail and the corepressor CtBP in repressing neuroectoderm genes (Nibu et al, 1998). The histone deacetylase HDAC3 is also involved in Snail-mediated silencing during ZGA, through cooperation with the Ebi/

SMRTER corepressor complex (Qi et al, 2008). HDAC3 corepression functions are independent of its catalytic activity, whereas its deacetylase activity is required for gene activation of other targets (Tang et al, 2023). In turn, CtBP cooperates with other early zygotic transcription factors in establishing nuclear identity, including Knirps, Krüppel, Giant and Hairy (Keller et al, 2000; Nibu and Levine, 2001; Nibu et al, 1998; Poortinga et al, 1998; Struffi et al, 2004; Strunk et al, 2001). Both HDAC3 and CtBP corepressors were also recently found as positive hits in a genetic screen for chromatin factors involved in early embryogenesis, confirming their importance during major ZGA (Ciabrelli et al, 2023).

Finally, the transcriptional corepressor Grunge is required for the correct segmentation of the embryo by repressing *hunchback*, *Krüppel*, *knirps*, and *fushi-tarazu* expression (Erkner et al, 2002), as well as its interacting partner Brakeless (Haecker et al, 2007) and also the corepressor Groucho, limiting the expression of *tailless* and *huckebein* to the poles of the embryo (Goldstein et al, 1999). Both Grunge and Groucho requirement for embryonic development was recently validated by a genetic screen (Ciabrelli et al, 2023), with the lack of Groucho having more drastic effects on development, even before major ZGA. In summary, different types of transcriptional corepressors act synergistically to spatiotemporally control gene expression at a time when thousands of genes are de novo transcribed. Their precise regulation is of paramount importance for embryonic development.

### Chromatin-mediated transcriptional activation

Chromatin complexes cooperate with pioneer factors and with specific transcriptional activators during ZGA. Both housekeeping and developmentally-regulated genes need to be de novo activated either in every nucleus or in specific embryonic domains, respectively. Different types of chromatin coactivators regulate these distinct classes of genes during ZGA. For instance, the HAT Nejire (twisted in Japanese) plays a major role in the regulation of hundreds of developmental genes (Ciabrelli et al, 2023). This massive transcriptional coactivator (332 kDa) (Akimaru et al, 1997) was initially characterized as responsible for the activation of early zygotic genes in flies (Akimaru et al, 1997). Mutations in Nej caused early lethality and a "twisted" embryo phenotype, originally linked to the downregulation of the *twist* gene (Akimaru et al, 1997), but actually due to defects in Dpp-signaling (Lilja et al, 2003). In order to activate *twist*, Nej acts as a coactivator of the transcription factor Dorsal, which in turn cooperates with Zld (Boija and Mannervik, 2016). In line with these observations, hundreds of Zld-dependent genes are regulated by Nej during ZGA (Ciabrelli et al, 2023). The Nej protein is responsible for the deposition of H3K18ac, H3K27ac and for H4 acetylations on K5 and K8 residues (Feller et al, 2015). Therefore, its contribution as a transcriptional coactivator has been historically connected to the deposition of these histone marks. Nonetheless, early reports showed a Nej catalytically independent function in the activation of pair-rule genes as *even-skipped* (Ludlam et al, 2002) or of the TGF-β signaling genes *rhomboid* and *tolloid* (Lilja et al, 2007). More recently, studies performed with the histone replacement system demonstrated that precluding acetylation of H3K27 alone (Pengelly et al, 2013) or together with H3.3K27 (Leatham-Jensen et al, 2019), does not affect active genes in flies as well as in mouse ESCs (Sankar et al, 2022; Zhang et al, 2019). In these studies, all observed phenotypes were compatible with the loss of H3K27me3 Polycomb-

mediated repression but not with failure in gene activation. Finally, recent work from our group demonstrated that loss of ZGA and embryonic viability upon maternal knockdown of Nej are rescued by maternal expression of a catalytically dead Nej protein (Ciabrelli et al, 2023), confirming the existence of Nejire non-enzymatic functions (Hunt et al, 2022). The specific point mutation we introduced to generate Nej catalytically dead enzyme allowed us to deactivate Nej's catalytic activity without affecting its ability to bind to chromatin or to interact with other essential partners. Maintaining the integrity of these interactions is essential, as any alteration could misrepresent the enzyme's function and complicate the interpretation of experimental results. These in vivo observations were further confirmed by in vitro experiments, showing that the lack of Nej catalytic activity only marginally decreased its transactivating function. Instead, deletion of the Nej NTD, which harbors the KIX domain, almost completely abolishes its transactivation function (Ciabrelli et al, 2023). The KIX domain in mammals is necessary for the formation of heterodimers with a range of transcription factors (Goto et al, 2002; Parker et al, 1996). Collectively, these studies seem to suggest that the Nej protein itself rather than its catalytic activity, could be crucial for gene regulation and embryonic development. How Nej regulates RNA PolII activity is still an open question.

The Gcn5 protein, a HAT responsible for H3K9ac and H3K14ac deposition, is crucial for ZGA. Maternal knockdown of Gcn5 (Ciabrelli et al, 2023) or of core subunits of the Gcn5-containing SAGA and ATAC complexes (Helmlinger et al, 2021; Li et al, 2017) affects cellularization and later stages of embryonic development. Lack of these components leads to the misexpression of hundreds of genes during the major wave of ZGA (Ciabrelli et al, 2023) and in later stages of embryonic development (Torres-Zelada et al, 2022). Unlike Nej, depletion of Gcn5 mostly affects housekeeping rather than developmental genes. Moreover, most of the Gcn5-dependent genes are not regulated by Zld and GAF at ZGA (Ciabrelli et al, 2023). Yet, both Nej-dependent (e.g., H3K27ac) and Gcn5-dependent acetylation marks (e.g., H3K9ac) are deposited on every active TSS during ZGA (Ciabrelli et al, 2023). This apparent discrepancy raises questions about the actual function of these histone acetylation marks. Indeed, despite the depletion of maternally provided Gcn5 before the end of embryogenesis, mutant flies also lacking the zygotic Gcn5 HAT activity can still complete several metamorphoses without any H3K9 acetylation and survive till the puparium stage (Carré et al, 2005). Furthermore, a mutation in the SAGA subunit Saf6 results in defective expression of SAGA-dependent genes, even though H3K9ac is not affected (Weake et al, 2009). These results show that H3K9ac is neither necessary nor sufficient to promote Gcn5-dependent gene activation in flies. Recently, our laboratory showed how a catalytically dead version of Gcn5 can completely rescue the Gcn5 mutant phenotype during early embryogenesis and ZGA. Our work demonstrated that the presence of Gcn5 itself, and therefore of its associated macromolecular complexes, but not of its catalytic activity, is important for the activation of hundreds of genes during ZGA and consequently for embryonic viability (Ciabrelli et al, 2023).

Other HATs also exert important roles during ZGA, even though they have been investigated to a lesser extent. For instance, the HAT Enok is responsible for H3K23ac deposition in vitro (Feller et al, 2015) and also during early embryogenesis (Huang et al, 2014). Although most of the genome seems to be occupied by

H3K23ac in cell culture conditions (Feller et al, 2015), only a few genes are affected by its absence during ZGA in living embryos (Ciabrelli et al, 2023; Huang et al, 2014). These results could be explained by a different distribution of the H3K23ac mark between cell culture conditions and ZGA. Alternatively, it is possible that the H3K23ac mark is also relatively abundant during ZGA, but the role of Enok during embryogenesis might not be strictly linked to transcription. For instance, this HAT might elicit PCNA unloading during DNA replication (Huang et al, 2016). Similarly, the functions of the Tip60 and the Chameau HATs, which are responsible for the acetylation of H4K12ac (Feller et al, 2015), have not been thoroughly investigated in the context of ZGA. Recently, we showed that the lack of Chameau results in morphological defects during early embryogenesis, with very few embryos reaching the gastrulation stage (Ciabrelli et al, 2023).

An evolutionarily conserved class of chromatin factors that are critical for gene activation during Drosophila ZGA are the Trithorax (Trx) group proteins (Kassis et al, 2017). Trithorax group genes are genetically classified as positive regulators of *Drosophila* Hox gene expression, which specify segment identity along the antero-posterior embryonic axis (Ingham, 1981). In accordance, Trx mutants phenocopy loss-of-function mutations in the Hox genes (Digan et al, 1986; Forquignon, 1981; Shearn et al, 1987). The founder of the Trx group is the *trx* gene itself, which encodes a histone methyltransferase responsible for H3K4 methylation. Besides Trx, flies express two other histone methyltransferases responsible for H3K4 methylation, namely Trithorax-related (Trr) and Set1, all belonging to the evolutionary conserved COMPASS complexes (Piunti and Shilatifard, 2016).

Trx catalyzes H3K4me1 on Polycomb/Trithorax regulatory elements (PRE/TRE) (Tie et al, 2014), although in vitro studies revealed that Trx mutations also affect H3K4me2 levels (Rickels et al, 2016). Set1 is responsible for the majority of the H3K4me2 and H3K4me3 deposition at TSSs in flies (Herz et al, 2012). The levels of H3K4me1 and H3K4me3 chromatin occupancy are very limited in *Drosophila* before major ZGA (Chen et al, 2013; Li et al, 2014; Samata et al, 2020), albeit a discrete number of H3K4me1 peaks could be already detected at nuclear cycle 8 (Li et al, 2014).

Trr is specifically responsible for H3K4me1 at enhancer regions, and its absence hinders H3K27ac (Calo and Wysocka, 2013; Piunti and Shilatifard, 2016; Rickels et al, 2017). It has been previously postulated that the deposition of H3K4me1 is a key determinant for enhancer functions (Creyghton et al, 2010). Nonetheless, flies bearing a SET-domain deficient Trr protein are viable, indicating that gene expression programs can be activated in the absence of H3K4me1, and despite reduced H3K27ac levels at enhancers (Rickels et al, 2017; Rickels et al, 2020). These findings were recapitulated in mammalian systems (Cao et al, 2018; Dorighi et al, 2017) and confirmed previous studies in flies (Hödl and Basler, 2012) and in yeast (Krogan et al, 2002; Nislow et al, 1997), showing that H3K4 methylation is not required for, but rather a consequence of, transcriptional activity (Morgan and Shilatifard, 2020). Other Trx group proteins operate at enhancers, including two components of the mediator complex (i.e., Skuld and Kohtalo), which promotes enhancer-promoter looping (Janody et al, 2003). Several components of chromatin-remodeling complexes were also identified as Trx group proteins, e.g., Brahma, the catalytic subunit of SWI/SNF complex (Mohrmann et al, 2004). Finally, the pioneer factor GAF, encoded by the *trithorax-like* gene, is also classified as a Trx group gene (Farkas et al, 1994). In conclusion, Trx group proteins are important regulators of ZGA genes, particularly those involved in developmental programs and expressed in specific regions, segments or presumptive germ layers.

### Chromatin accessibility

At ZGA, anterior-posterior and dorsal-ventral axes have already been established, and differential gene expression is a direct consequence of embryo axial polarization (Campos-Ortega and Hartenstein, 1985). Does chromatin accessibility mirror such a variegated transcriptional pattern? Chromatin accessibility is similar in the anterior and posterior halves of nuclear cycle 14 embryos (Haines and Eisen, 2018). Therefore, based solely on accessibility, it is not possible to distinguish between anterior and posterior embryos. In the future, assessing the accessibility of individual nuclei will be necessary to address this question. Remarkably, even the promoters of those genes that are expressed in the anterior half of the embryo do not display different levels of chromatin accessibility along the AP axis (Haines and Eisen, 2018). Interestingly, the only exceptions are enhancer regions, which show a mild increase in chromatin opening in the half of the embryo where they are supposed to operate, even though partial chromatin opening is also observed in the opposite and inactive half (Haines and Eisen, 2018).

Nuclear sorting of specific segments along the AP axis through an exogenous reporter system allowed for a more fine-grained resolution of the chromatin landscape during *Drosophila* blastoderm stage (Bozek et al, 2019). This work confirmed that the vast majority of open-chromatin peaks are conserved in different regions of the embryo at bulk but not at single nucleus resolution. Moreover, differences in chromatin accessibility in one-quarter of the detected peaks were enriched at enhancers regions (Bozek et al, 2019). Chromatin accessibility was also investigated in DV tissue mutant embryos composed of presumptive dorsal ectoderm, neurogenic ectoderm, or mesoderm. In line with results obtained for the AP axis, enhancers showed increased accessibility along the DV axis (Hunt et al, 2024).

Deciphering chromatin accessibility differences in the heterogeneous Drosophila embryo is a complex task. The advent of single-cell techniques further shed light on the chromatin landscape across early embryogenesis. Single-cell ATAC-seq experiments corroborated the observations collected in earlier bulk studies using the DNAse I enzyme (Thomas et al, 2011). The results showed high variability in chromatin accessibility among different developmental windows throughout embryogenesis (Calderon et al, 2022; Cusanovich et al, 2018). Moreover, nuclear heterogeneity in chromatin accessibility along the embryo's AP axes was confirmed to be a feature of early embryos, in particular at enhancer regions (Haines and Eisen, 2018).

Single-cell chromatin accessibility confirmed this pattern not only for the AP axis, but also for the DV axes (Cusanovich et al, 2018). Interestingly, some studies showed that the presence of transcription factor binding sites alone could not completely explain these differences in chromatin accessibility during regionalization (Haines and Eisen, 2018; Hannon et al, 2017). Due to the intricate nature of early *Drosophila* development, capturing precisely the chromatin dynamics during ZGA has always been challenging and despite many efforts in the field, a detailed map at

single-cell resolution of the chromatin landscape at this stage is still missing. Moreover, studies aimed at understanding the surrounding chromatin context of regulatory elements may be crucial to elucidate the mechanisms shaping the chromatin dynamics of early embryogenesis.

Chromatin accessibility at this stage is also achieved by the action of pioneer factors. Their main function consists of exposing enhancers' chromatin to the binding of transcription factors. The activation of about 600 genes at nuclear cycle 14 is dependent on Zld (Ibarra-Morales et al, 2021; Liang et al, 2008), whereas about 400 genes directly depend on GAF (Gaskill et al, 2023). Clamp mostly cooperates with Zld in the expression of early ZGA genes (Duan et al, 2021) but also acts independently on other targets (Colonnetta et al, 2021). Other genes rely on multiple pioneer factors, and this redundancy ensures that the lack of one factor will not decisively compromise their activity (Colonnetta et al, 2021; Duan et al, 2021; Gaskill et al, 2021).

Whereas many Zld and Clamp-dependent genes are expressed early during cleavage, GAF-dependent genes are active only at nuclear cycle 14 (Blythe and Wieschaus, 2016; Gaskill et al, 2021). At this stage, hundreds of genes are regulated at the RNA PolII pausing/elongation step (Blythe and Wieschaus, 2016; Saunders et al, 2013), in order to achieve coordinated gene expression (Lagha et al, 2013). Indeed, shorter cell cycle lengths, which are characteristic of early ZGA stages, are not compatible with this type of transcriptional regulation (Chen et al, 2013; Kwasnieski et al, 2019). The idea that GAF could be the main factor involved in the regulation of paused cycle 14 genes is corroborated by in vitro studies in *Drosophila* cell lines. These studies show that GAF-dependent genes are indeed regulated at the RNA PolII pause/release and transition to the productive elongation step (Boija et al, 2017). In addition, GAF-dependent genes are enriched for core promoter elements typically found at pause/release regulated genes, whereas Zelda-dependent genes are instead enriched with the TATA-box motif, typical of genes regulated at the RNA PolII recruitment step (Chen et al, 2013). Nonetheless, ZGA genes that are coregulated by Zelda and GAF, are also regulated at the RNA PolII pause/release step, indicating that Zelda-mediated regulation is also compatible with this molecular mechanism (Boija and Mannervik, 2016).

How do pioneer factors manage to render chromatin accessible and transcriptionally competent? Are chromatin-remodeling complexes recruited by pioneer factors to evict or displace nucleosomes during ZGA?

By catalyzing ATP-dependent alterations in nucleosome structure or positioning, chromatin-remodeling complexes regulate chromatin accessibility. Notably, several components of chromatin remodelers represent an additional class of Trx group proteins, including the ATPase subunit of the SWI/SNF complex Brahma and other SWI/SNF components, Moira and Osa. These proteins are part of the Brahma-associated protein complex (BAP) and Polybromo-containing BAP complex (PBAP) (Mohrmann et al, 2004). Mutations in these genes cause homeotic transformations (Harding et al, 1995; Tamkun et al, 1992). Certain loss-of-function mutants (e.g., *osa*) phenocopied gap gene mutants, showing that these chromatin-remodeling complexes are already involved in the very early expression of zygotic genes regulating AP patterning (Vázquez et al, 1999).

ATP-dependent chromatin-remodeling complexes represent the most abundant class of proteins acting on chromatin at ZGA, suggesting that they also have a structural role (Bonnet et al, 2019). Studies performed in *Drosophila* cell cultures revealed that GAF functions with PBAP (SWI/SNF family) to open up chromatin. Downstream of this step, GAF synergizes with NURF (ISWI family) in order to ensure efficient RNA PolII pause/release and transition toward positive elongation (Judd et al, 2021). It is possible that GAF acts similarly also during ZGA, but this hypothesis needs to be experimentally tested.

Could Zld also cooperate with chromatin-remodeling factors, or is its function independent of them? Future work is required to elucidate the interplay between pioneer factors and chromatin remodeling complexes during ZGA. Even though pioneer factors might cooperate with chromatin-remodeling complexes to activate developmental genes, it is important to note that most major ZGA genes are pioneer-factor independent. Their expression instead relies on the deposition of the histone variant H2Av at their +1 nucleosome mediated by the histone chaperone Domino (Ibarra-Morales et al, 2021) (Fig. 3, top). As expected, lack of maternally deposited Domino causes embryonic lethality. H2Av-dependent genes display nucleosome phasing, unlike Zld-dependent genes, whose nucleosomes are devoid of histone H2Av. Interestingly, however, H2Av deposition per se does not affect chromatin accessibility or nucleosome positioning at these loci. Therefore, nucleosome distribution must be dictated by other chromatin factors (Ibarra-Morales et al, 2021). How the presence of H2Av mechanistically contributes to RNA PolII activity for thousands of genes at ZGA is still unclear.

### Chromatin architecture

Chromatin architecture acquires distinctive features during the major wave of ZGA. Randomly folded chromosomes begin to organize inside the nuclear space in concert with ZGA. At nuclear cycle 14, TADs are finally well-defined (Gizzi et al, 2019; Hug et al, 2017; Ogiyama et al, 2018) (Fig. 3, bottom), and their internal organization changes considerably upon transcriptional activation (Gizzi et al, 2019). The temporal correlation between increased RNA PolII activity and TAD establishment does not bear a causal link. Indeed, chemical inhibition of the RNA PolII activity does not interfere with TAD formation (Hug et al, 2017). Thus, TAD establishment is independent of transcriptional onset. Nevertheless, transcriptional inhibition results in a mild loss of inter-TAD insulation, with increased inter-TAD and decreased intra-TAD interactions (Hug et al, 2017).

Housekeeping genes are enriched at TAD borders, and they tend to cluster three-dimensionally with other housekeeping genes (Hug et al, 2017). In addition to histone marks, insulator-binding proteins, such as Beaf-32, CP190, and CTCF, also reside at TAD borders (Sexton et al, 2012) (Fig. 3, bottom). Despite their suggestive subnuclear distribution, depletion of the factors during early embryogenesis does not interfere with TAD establishment, and ZGA is also mostly unperturbed (Cavalheiro et al, 2023). In accordance, CTCF and CP190 are not required for the completion of *Drosophila* embryogenesis, and their impact on gene expression is not predominant (Gambetta and Furlong, 2018; Kaushal et al, 2022). Nonetheless, the simultaneous presence of at least one active promoter and an insulator-bound region is required for the full

insulation of a TAD border (Cavalheiro et al, 2023). The position of TAD borders and other features of chromatin conformation are surprisingly conserved during early embryonic development (Ing-Simmons et al, 2021). Indeed, compartments, TADs, and even enhancer-promoter loops are highly similar across presumptive germ layers in the Drosophila cellular blastoderm nuclei (Espinola et al, 2021; Ing-Simmons et al, 2021).

Besides their presence at TADs, insulator-binding proteins are also enriched at tethering elements. Tethering elements can be defined as genomic sequences that allow nearby enhancers to engage in long-range interactions with promoters. TAD boundaries and tethering elements have distinct roles and properties, even though they share some similarities. TAD boundaries prevent spurious interactions between enhancers and silencers harbored in distinct TADs, thus compartmentalizing de facto the genome in functional units. Tethering elements instead ensure precision and timing of transcriptional dynamics (Batut et al, 2022). Consistent with this, disruption of TAD boundaries causes TAD fusion, whereas disruption of the tethering elements affects enhancer function or its precision, with minimal impact on TAD structure. Interestingly, multiple enhancers, driven by their tethering elements, physically interact to form hubs, which usually contact only one promoter at a time during ZGA (Espinola et al, 2021). During ZGA, TAD boundaries are enriched for CTCF, Cp190, and H3K4me3, whereas tethering elements are enriched in pioneer factors and H3K4me1 (Batut et al, 2022). After ZGA, Polycomb-repressed regions form repressive loops, relying on the action of GAF, which is bound at PREs (Ogiyama et al, 2018). The ability of PRC1 to multimerize via the oligomerization of Ph and Scm SAM domains (Isono et al, 2013) can stabilize these GAF-dependent long-range interactions (Loubiere et al, 2020). The clustering of Polycomb-repressed chromatin domains inside the nucleus results in the formation of the so-called "Polycomb bodies" (Bantignies et al, 2011; Lanzuolo et al, 2007). GAF can also mediate long-range interactions between active promoters through its POZ/BTB multimerization domain (Li et al, 2023).

Constitutive heterochromatin regions coalesce during *Drosophila* ZGA. Hi-C experiments (Lee et al, 2020; Zenk et al, 2021) have confirmed previous studies showing that pericentromeric regions of different chromosomes are spatially connected in a chromocenter (Miklos and Cotsell, 1990). In blastoderm nuclei, centromeres start to cluster at the apical pole, in the so-called "Rabl conformation" (Foe et al, 2000; Marshall et al, 1996). Telomeres also coalesce at the nuclear periphery but in separate regions (Lee et al, 2020). The strongest heterochromatic interactions reside within the same chromosome arm, according to genomic partitioning in chromosome territories (Lee et al, 2020). Moreover, H3K9me2 regions embedded in chromosome arms, including transposable elements, show preferential interactions with the apical chromocenter, where pericentromeric heterochromatin resides (Lee et al, 2020).

Recently, our laboratory showed that HP1a binding at pericentromeric regions is necessary to establish constitutive heterochromatin clustering. HP1a depletion specifically affects the inactive B compartment and culminates in the loss of contacts within and between pericentromeric regions. Lack of H3K9me2/3 obtained with H3K9M mutant embryos also resulted in pericentromeric heterochromatin de-clustering (Zenk et al, 2021). Interestingly, HP1a at ZGA localizes at constitutive heterochromatin,

but one-third of HP1a peaks are detected on chromosome arms, where it is responsible for correct chromosome folding. In agreement with this, HP1a depletion reduces inter-chromosomal contacts while increasing intrachromosomal contacts and impairing the proper segregation of the A and B compartments. Despite its important role in chromatin organization, the lack of HP1a affects the expression of just a handful of genes and repeats at ZGA. Therefore, HP1a's role at this particular developmental stage is mostly different from its role in differentiated somatic cells (Zenk et al, 2021). Indeed, in differentiated, somatic *Drosophila* S2 cells, HP1a depletion does not lead to significant changes in genome architecture (Zenk et al, 2021). Moreover, in mouse fibroblast cells, the size, accessibility, and compaction of pericentromeric heterochromatin foci are independent of HP1 (Erdel et al, 2020). In the same cells, HP1 exhibits a very low capacity to form liquid-liquid phase-separated droplets (Erdel et al, 2020). Therefore, we speculate that the function of HP1 in driving B-compartment establishment might rest on the ability of HP1a to assemble constitutive heterochromatin domains specifically in early fly embryos (Atinbayeva et al, 2024; Strom et al, 2017). Finally, in line with previous studies (Fung et al, 1998), the level of homologous chromosome pairing drastically increased from nuclear cycle 12 to nuclear cycle 14 embryos, reaching the pairing frequency of around 70% of the total at nuclear cycle 14 (Gizzi et al, 2019; Hiraoka et al, 1993) (Fig. 3, bottom).

# Intergenerational and transgenerational inheritance

Maternally inherited chromatin is not a blank page. Key epigenetic marks such as H3K27me3, H3K9me3 and H4K16ac already decorate the epigenome at specific loci in the zygote (Atinbayeva et al, 2024; Samata et al, 2020; Zenk et al, 2017). Experimental perturbation of this intergenerational inheritance leads to defects in Polycomb-mediated silencing, heterochromatin activation, and chromatin accessibility at promoters of late-expressed genes during embryogenesis. Whereas most of the tested histone modifications are not intergenerationally inherited (e.g., H3K4me1, H2K27ac, H3K9ac), other histone marks might be retained in unfertilized eggs. Interestingly, the intergenerationally maintained histone marks characterized so far (i.e., H3K27me3, H3K9me3 and H4K16ac) are the ones with a proven "instructive" function on the underlying genetic material (Copur et al, 2018; Leatham-Jensen et al, 2019; Pengelly et al, 2013; Penke et al, 2016). This function might explain why they are intergenerationally transmitted, whereas other marks that are rather a consequence of transcriptional activity or other chromatin processes are instead not transmitted. Because protamine replacement does not happen on the entire paternal epigenome in *Drosophila* (Elnfati et al, 2016), it is tempting to speculate that modified histones might also be paternally transmitted to the progeny as they are in mouse (Brunner et al, 2014; van der Heijden et al, 2006). It is also possible that other epigenetic marks besides histone modifications can be passed on from the father to the offspring. For instance, since protamines themselves can be post-translationally modified, they might be novel carriers of epigenetic information.

Epigenetic inheritance of these marks may convey critical information for the correct development of the next generation

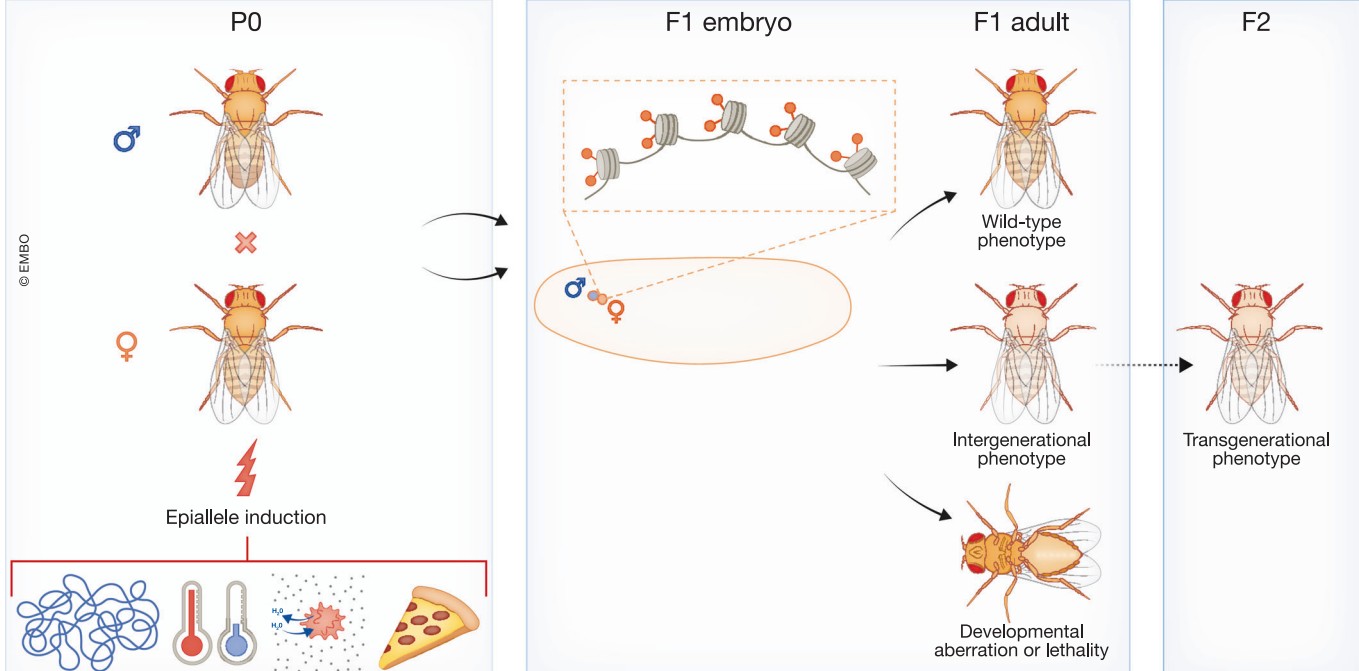

**Figure 4. Intergenerationally inherited chromatin marks, epiallele induction and transgenerational phenotypes.**

Left, parental (P0) generation. Parents can transmit their repertoire of histone modifications (in red) in physiological conditions, or after epiallele induction (e.g., chromatin interactions, temperature stress, osmotic stress, diet stress) to the next generation (right). The F1 embryo could inherit a standard set of histone modifications or, alternatively, a modified one, which could impair fly development or produce a viable intergenerational phenotype in the F1 adult. If the phenotype persists in the following (i.e., F2) generations, it is defined as transgenerational epigenetic inheritance.

(Fig. 4). On the other hand, they may carry more flexible and complementary types of information that might be generated by the response to specific stimuli in the parental lifespan. Intergenerational inheritance refers to the epigenetic information transmitted from the parent to the offspring. When the epigenetic information is stably maintained across multiple generations, it is referred to as transgenerational epigenetic inheritance (TEI) (Fitz-James and Cavalli, 2022) (Fig. 4). Depending on the experimental system, TEI can originate from both parents (Ciabrelli et al, 2017; Seong et al, 2011) or from the mother only (Bozler et al, 2019; Cavalli and Paro, 1998). Interestingly, early embryonic development seems to be a developmental hotspot for TEI induction in flies (Cavalli and Paro, 1998; Seong et al, 2011), suggesting that at this stage, a plastic chromatin state can integrate and propagate epigenetic signals. Strikingly, TEI of alternative Polycomb-mediated repression has been described in flies when induced by increased frequency of chromatin long-range interactions (Bantignies et al, 2003; Ciabrelli et al, 2017). TEI is also triggered when fly larvae are exposed to specific antibiotics, resulting in ectopic induction of gene expression and longer pupation time in the following generations (Stern et al, 2012; Stern et al, 2014). Constitutive heterochromatin formation can be perturbed by the exposure of previous generations to environmental stress. Environmental factors can also hinder constitutive heterochromatin formation in a transgenerationally inheritable fashion. Indeed, heat shock or osmotic stress induces the phosphorylation of ATF-2, its consequent release from constitutive heterochromatin, and failure in H3K9me2 deposition (Seong et al, 2011). This disruption can be inherited by the next

generations despite the removal of the environmental stimulus. Whether the H3K9me3 mark itself, a noncoding RNA species, or alternative molecular carriers are responsible for this type of inheritance is still an open question.

Finally, TEI of active states has also been described in flies. For instance, heat-shock-mediated Gal-4 transactivation during embryogenesis can be mitotically inherited until the adult stage and meiotically inherited from the maternal side by the following generations (Cavalli and Paro, 1998). On the other hand, there are several other TEI studies in flies that lack a thorough molecular characterization. Many of these studies seem to lend support to a chromatin-based mechanism for epiallele inheritance, and it would be of great interest to test this hypothesis in the future. For instance, several reports point to diet-induced intergenerationally or transgenerationally inherited phenotypes. High dosage of sugar in paternal (Öst et al, 2014) or maternal (Buescher et al, 2013) diet induces obesity in the F1 or even the F2 generation, respectively. Furthermore, an imbalance in protein to carbohydrate ratio (Towarnicki et al, 2022; Xia and de Belle, 2016) or genetically induced transient obesity (Palu et al, 2017) can have deleterious effects on the offspring. Strikingly, an imbalance in H3K27me3 and H3K9me3 in the offspring during specific developmental windows has already been observed when the parents were exposed to diet perturbation (Öst et al, 2014). Finally, exposure of flies to Cadmium (Sun et al, 2023) or to predatory wasps (Bozler et al, 2019) was shown to induce transgenerationally inheritable phenotypes such as wing defects or ethanol preference, respectively. Intriguingly, Cadmium exposure results in global changes of histone marks

such as H3K9me3 and H3K27me3 in the exposed generation, which are both intergenerationally inherited (Atinbayeva et al, 2024; Zenk et al, 2017). Differently, the inheritance of ethanol preference correlates with the inheritance of the maternal epigenetic state of a single locus encoding the Neuropeptide-F (Bozler et al, 2019).

## Discussion

Chromatin fibers are subjected to enormous changes in the short window of time between fertilization and cellularization in *Drosophila* embryos. In these first 3 hours of fly development, the chromatin reorganizes, and the epigenome integrates all the signals required for accurate gene activation or repression. The chromatin transitions from random folding into a highly organized structure, which allows the genetic material to exert its main functions. Tight regulation of these processes is paramount for precisely executing the fly developmental program.

A growing body of evidence supports a model whereby intergenerationally inherited chromatin is not transmitted in a naïve state, but is already associated with histone marks such as H3K9me3 and H3K27me3 deposited at specific loci. Once inherited, these marks must be propagated across mitotic divisions in order to maintain the epigenetic information on specific loci. The mechanisms in place in Drosophila early embryos to propagate certain epigenetic states might be, in principle, the same as observed in other systems. During S phase, a reading mechanism is involved in the recognition of the epigenetic mark and the corresponding enzyme is triggered to catalyze the same modifications on the newly assembled nucleosome (Margueron et al, 2009; Ragunathan et al, 2015). It is possible however that additional mechanisms may be in place to maintain the intergenerationally inherited information. Given the smaller size of the inherited H3K27me3 and H3K9me3 domains in early embryos compared to their sizes at later developmental stages, higher precision may be required to transmit these marks during the early stages.

The presence of these histone modifications is important for canonical developmental programs through constitutive heterochromatin formation (Atinbayeva et al, 2024), Polycomb-mediated gene silencing (Zenk et al, 2017), and transcriptional activation of housekeeping genes (Samata et al, 2020). The rapid increase of these marks at the onset of ZGA depends on their specific functions. For instance, H3K27me3 role is to silence thousands of genes that, at cycle 14, are now ready to be transcribed. Similarly, H3K9me2/3 levels increase toward ZGA, and play an essential role for the proper centromeric/pericentromeric compaction and nuclear division (Atinbayeva et al, 2024). Together with hardwired developmental programs, a certain degree of epigenetic flexibility on specific loci could determine the inheritance of alternative epigenetic states across generations. We hypothesize that H3K9me3 and H3K27me3 (Atinbayeva et al, 2024; Zenk et al, 2017), in addition to their physiological roles in epigenetic inheritance, might also transmit environmentally induced changes in germline chromatin. Future research in this direction is necessary to fill the gap between an increasing number of reports of chromatin-associated TEI phenomena and the molecular mechanisms underlying their inheritance.

---

**Box 1    In need of answers**

How are A-compartments established and maintained in the early embryo?

How are long-range loops established and maintained in the early embryo?

What role does CBP/Nejire play in the activation of ZGA?

How are chromatin remodelers recruited to chromatin right after fertilization? Is this via pioneer factors or other mechanisms?

How are Polycomb domains established on the paternal chromosomes after fertilization?

How are tethering elements stabilized in the early embryo?

What is the timing of enhancer RNA expression? Are they expressed before or concomitantly with the ZGA?

Who are the recruiters of E(z), Su(var)3-9 and dSetDB1 complexes?

How HP1a is recruited to the chromatin independent of H3K9 methylation in the early embryos?

Do maternal H3K9me3 and H3K27me3 have functions in early meiosis?

How do HP1a foci behave in single mutants of H3K9 histone methyltransferase?

How, when, and where are HDACs recruited in the early embryo?

---

In conclusion, while significant progress has been made in understanding the chromatin dynamics of early embryos, our knowledge is far from complete. As we have outlined in Box 1, numerous critical questions remain unanswered, underscoring the vast potential for future research in this field. The journey toward discoveries is ongoing, and unraveling these mysteries will be key to advancing our comprehension of early embryonic development.

## Peer review information

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

## Acknowledgements

The authors are very thankful to the Iovino lab members, particularly to Francesco Cardamone for his feedback on the manuscript and the rest of the current and previous Iovino lab members for their constant support. NI is supported from the Max Planck Society; CIBSS–EXC 2189; Deutsche Forschungsgemeinschaft–Project ID 192904750–CRC 992 Medical Epigenetics and European Research Council (ERC) under the European Union's Horizon 2020 research and innovation programme (grant agreement No. 819941) ERC CoG, EpiRIME.

## Author contributions

**Filippo Ciabrelli**: Conceptualization; Visualization; Writing—original draft; Writing—review and editing. **Nazerke Atinbayeva**: Visualization; Writing—original draft; Writing—review and editing. **Attilio Pane**: Writing—original draft; Writing—review and editing. **Nicola Iovino**: Conceptualization; Resources; Supervision; Funding acquisition; Visualization; Writing—original draft; Project administration; Writing—review and editing.

## Funding

## Disclosure and competing interests statement

The authors declare no competing interests.

