## [Peer Review File · EMBO Reports]

Epigenetic Inheritance and Gene Expression Regulation in Early *Drosophila* Embryos

Nicola Iovino, Filippo Ciabrelli, Nazerke Atinbayeva, and Attilio Pane

Corresponding author(s): Nicola Iovino (iovino@ie-freiburg.mpg.de)

Review Timeline:

Submission Date:	22nd Feb 24
Editorial Decision:	15th Mar 24
Revision Received:	13th May 24
Accepted:	21st Aug 24

Editor: Esther Schnapp

Transaction Report:

Dear Nicola,

Thank you for the submission of your Review to EMBO reports. We have now received the full set of referee reports on it, pasted below.

As you will see, all referees acknowledge that the review is timely and interesting, which is great. They also have several suggestions for how the review could be further improved, and I think all suggestions are good and should be addressed. I am a bit less sure about referee 3's comments and leave it up to you how to address these. It would be very helpful if you could address all comments in a point-by-point response that I would like to ask you to co-submit with your final ms to ease ms evaluation.

I agree with referees 1 and 3 that adding more of your personal point of views and interpretations of published data and your conclusions will make the review more interesting for the readers. It will also be important that you upload the unpublished data you discuss from your lab on BioRxiv, if these have not been published yet in a journal. It is also true that EMBO reports does not allow the statement "data not shown", which will need to be removed.

I would further like to suggest that you add one more figure to the review, if possible. We also ask our authors to include a Box in their reviews called "In need of answers" where major open questions in the field are listed. These can be accompanied by suggested experimentation addressing the questions. This Box might also help with addressing some of the referees comments.

As for timing, would it be possible for you to submit the revised review by April 7th? If you need more time, please just let me know.

Thank you again for writing this timely review for us!

With best wishes,
Esther

Referee #1:

Ciabrelli et al have reviewed the role of chromatin regulators and histone modifications in early embryonic development of Drosophila. It is a nice summary, but would benefit from a discussion about outstanding questions. For example, how histone marks are transmitted during the first rapid nuclear divisions, and why some marks appear to be intergenerationally inherited and others not.

Comments:

1. The authors have structured the review into 3 parts; Fertilization and early divisions, minor ZGA and major ZGA. Although I understand the thought behind this, to me it feels repetitive to discuss heterochromatin, chromatin architecture etc multiple times. The authors may consider a short outline of the timeline of events, followed by a discussion of each topic (chromatin mark inheritance, constitutive heterochromatin etc).
2. In the Chromatin mark inheritance section, the recent paper from the Loppin lab describing a role for the paternal loss protein in evicting H3-H4 tetramers from spermatid DNA should be mentioned (PMID: 37943933).
3. I could not understand why CID and Homothorax is discussed in the context of constitutive heterochromatin (p. 5, it is annoying that page and line numbers are missing from the ms).
4. Further down on same page, the section that starts with Histone replacement...pointed out that E(z) enzymatic activity... should be replaced with E(z) histone methyltransferase activity....
5. In this section Nejire is mentioned for the first time. For non-Drosophila readers, it would be beneficial to point out that this is the homolog of p300/CBP.
6. At the bottom of this page and top of next page, it is suggested that H3K27me3 is necessary to protect enhancers from Nej binding. However, this model is in conflict with evidence showing that Nej occupies many H3K27me3 regions (PMID: 36206738).
7. Top of p. 6 should also include the following reference (PMID: 25669886) when discussing the centrality of H3K27me3 in gene silencing.
8. The authors claim that histone modifications play instructive roles (e.g. H4K16ac on p. 6). What do they mean by this? Clearly, histone modification cannot direct cell-specific gene expression without transcription factors.
9. It would be interesting to read the authors' view on why H3K9me3 and H3K27me3 increase dramatically at the onset of ZGA.
10. I'm missing a discussion about how histone marks that are deposited on the maternal chromosome could be propagated

during the nuclear divisions that follow fertilization.

11. The description of HDAC3 on p. 14 is a little misleading. HDAC3 does not cooperate with CtBP. Instead, there are 2 co-repressor complexes involved in Snail-mediated repression, CtBP that interacts with HDAC1, and Ebi/SMRTER that interact with HDAC3. The authors may also want to mention that HDAC3 has both catalytic and non-catalytic roles in early embryo development (PMID: 37455638).

12. It is strange to mention the co-repressor Grunge without mentioning Brakeless. Furthermore, a really key co-repressor in the early embryo is Groucho that should also be included.

13. This is a minor point, but the twisted phenotype resulting from nejire mutations is not due to downregulation of twist as originally proposed, but to defects in Dpp-signaling (see e.g. PMID: 14550792).

14. When discussing TrxG proteins in gene activation (p. 16 and 17), Brm and other SWI/SNF components are missing, although they are discussed in a later section.

15. Differences in chromatin accessibility along the DV axis were recently described (PMID: 38166964), and could be contrasted to the data from the AP axis mentioned in the manuscript.

Referee #2:

Overall, this is a nice, extensively cited review focusing on how chromatin structure is reorganized during the maternal-to-zygotic transition in *Drosophila*. Despite numerous recent reviews on the MZT and this transition in *Drosophila* more specifically, the focus of this review on chromatin is distinctive and is an area that has been relatively less well documented than gene expression dynamics. As such, this review will be an important resource for many researchers.

One major concern with the review is the numerous discussions and 16 citations to Atinayeva et al. This appears to be an upcoming manuscript from the lovino lab but is not available online (either published at a peer reviewed journal or on bioRxiv). Thus, it is impossible to assess the data in this manuscript and how it impacts the current review. As such, it was not possible to review this aspect of the review. Similarly, the authors cite unpublished data from the lab. Again, this lack of transparency limits the ability of the reviewer to assess the data and therefore the conclusions that can be drawn. All the data discussed in a review should be available to the reviewer. That was not the case for this review.

Specific comments:

Figures were never referenced in the body of the manuscript. As such, it was challenging to determine how they were increased the clarity of the review.

It is confusing that the Introduction only covers a discussion of zygotic genome activation, despite the fact that heterochromatin establishment is extensively covered. The authors should consider mentioning the heterochromatin in the Introduction.

The mention that "the maternally provided pool of mRNAs inevitably declines" in Major ZGA; chromatin-mediated transcriptional silencing makes this sound like a passive process. It would be important to rephrase so that it is clear that ZGA is actively tied to this degradation.

In the discussion of mitotic retention, it could be useful to discuss the challenges posed by mitosis and the fact that early expression of the pioneer factor Zelda can't activate early gene expression (Larson et al. G3 2022).

Nej is first defined in Fertilization and early divisions; chromatin-mediated transcriptional silencing. But the reference is to the naming being related to twisted in Japanese is not until later. Perhaps reference this during the initial mention of the protein. In addition, a few lines below the reference to twisted "gene" is italicized after the gene twist when it should not be.

The Discussion rather than summarizing and bringing together the data brings up the entirely new concept of transgenerational inheritance of chromatin marks. This is an important area that the lovino lab has studied extensively. Perhaps this would be best to include as a stand-alone section prior to a Discussion.

Some of the text in the figures is too small to be legible.

In Figure 1: H3K27me3 and H3K9me3 levels appear close to maximum at stage 2. This is not what the data show. There is a clear increase in these marks at stage 5. This should be reflected in the figure. The authors may also want to indicate H1 levels.

In Figure 2: it could be good to label the male and female flies for a non-*Drosophila* audience.

While the authors should be commended for their well-cited manuscript, there are a selection of places that additional citations should be added:

Introduction:

Lott et al. PLoS Biol 2011 should be cited in the Introduction regarding the minor wave of zygotic genome activation.
Heyn et al Cell Rep 2014 should be cited for genes being short and lacking introns.
Soluri et al. eLife 2020 should be cited with Koromilla et al. for the role of Odd-paired.

Fertilization and Early Divisions; Chromatin-mediated transcriptional activation
Ali-Murthy et al. PLoS Genet 2013 should be cited regarding earliest observed transcription.
ten Bosch et al Development 2006 would be a useful citation regarding the timing of specific patterning and sex-determination genes.
Li et al. eLife 2014 should be cited along with Chen and Samata for histone modifications.

Major ZGA; Constitutive heterochromatin

Seller et al. Genes Dev 2019 should be cited for the role of histone methyltransferases in heterochromatin formation.

Major ZGA; chromatin-mediated transcriptional activation

Quite late in this section is a reference to Li, Harrison et al 2014. This seems to be incorrectly placed. It should be deleted and the appropriate reference included.

Hunt et al. Mol Cell 2022 should be mentioned to support the role of catalytic independent roles of Nej and the balance with H3K27me3.

Major ZGA: chromatin accessibility

It should be mentioned that promoters are generally accessible regardless of transcriptional state.

Liang et al. Nature 2008 should be included with Ibarra-Morales, Rauer et al 2021 for the ability of Zelda to broadly activate gene expression.

"Among them, the ATPase subunit of the SWI/SNF complex Brahma," is a sentence fragment.

There is a completely italicized sentence that should not be italicized in this section.

Major ZGA: chromatin architecture

It could be useful to initially explain tethering elements as they are relatively newly defined chromatin structures.

The role of GAF in establishing heterochromatin during this time should be discussed in relation to H3K9me3 and HP1a. Li et al. Mol Cell 2023 and Gaskill et al. Dev Cell 2023 would be good to discuss.

Referee #3:

The manuscript is an encyclopedic review of the fields of chromatin and transcription during early Drosophila embryogenesis. The authors cover all possible topics with an emphasis on the more recent literature, which is cited in a balanced manner. I think the review will be helpful and a reference for those interested in this topic. I went through the manuscript with a microscope and I have a lot of suggestions that the authors may think are too picky. Many of the suggestions, which correspond to questions elicited by my reading, may be a consequence of how the manuscript is written. I think the goal of the authors is to have everything known about the topic in one place. As a consequence, the manuscript reads as a collection of many facts linked to each other by the topic under discussion in each section. From my perspective, and I recognize that this may be my own biased view, the authors do not attempt to distil the information of what is known and attempt to gain a mechanistic understanding of what is happening during Drosophila embryogenesis. The understanding of the relationship between chromatin and transcription is much more detailed and sophisticated in mammals. The Drosophila field would benefit by attempting to explain what happens in Drosophila in the context of what is known in mammals. A second issue is that the authors take what has been published at face value, without critically evaluating whether the interpretation of things that were published 5 years ago is still correct, and without attempting to re-interpret older results in the context of what we know today in order to draw a more mechanistic understanding of the topic. The following is a list of things that popped in my mind while reading the manuscript:

1. "Its genome is largely dormant". Transcriptionally dormant may be a more appropriate term. The genome is replicating once every 8-10 min during this time, so it's busy doing other things.
2. Since the review is probably written for a large audience, not just Drosophila specialists, it may be better to avoid using specialized terms, like Bownes' stages. It would more intuitive to the general reader to refer to stages in terms of nuclear cycles.
3. The description of the replacement of protamines for histones in the male pronucleus is described as if the embryo specifically wants H3.3 to be present in the paternal but not the maternal chromosome. This histone H3.3 does not appear to play any

specific role in embryonic development based on the description, and it is eventually replaced by H3. I wonder if this is just a consequence of what is available in the zygote at the time. Are histone H3.3, ASF1, and HIRA the only ones present in the oocyte or are canonical histone proteins and their chaperones also present? The oocyte must have a lot of mRNAs encoding canonical histones. Do these RNAs have to be translated to provide histone proteins for the next S-phase or are the proteins already present? I imagine the oocyte nucleus is going through meiosis II, so it is not surprising that it keeps the histones it already had.

4. There is no reference in the list for work cited as Atinbayeva et al. When the authors state that "we demonstrated that H3K9me3 is not de novo established, rather it is intergenerationally inherited from the oocyte", is this another way of saying that the oocyte was in meiosis I, it was then fertilized, it now goes through meiosis II, and it kept the H3K9me3 it had before? In the discussion of heterochromatin and H3K9me2/3, authors should keep in mind work for the Bas van Steensel's lab showing that these covalent modifications are present at actively transcribed genes in what is normally considered to be euchromatin.

5. It would be interesting to consider what is happening to heterochromatin in the early embryo in more detail. Does the formation of heterochromatin depend on the formation of biomolecular condensates? If so, what does it mean that HP1 is absent from regions containing H3K9me3? The authors consider H3K9me2 and H3K9me3 as being equivalent, but regions containing H3K9me2 do not interact with regions of H3K9me3, suggesting that they may not participate in the same biomolecular condensates. Do the two modifications overlap in ChIP-seq results using antibodies specific to each one?

6. The review describes the standard view of transcription in the Drosophila embryo i.e., that something special takes place at the ZGA and pioneer factors decide it's time to act. However, an equally plausible scenario is that everything is in place to start transcription in the zygote (see, for example, PMID: 23593026) but transcription does not happen because the short cell cycles do not allow for a sufficiently long interphase to complete transcription. In this view, there is nothing special about ZGA other than the fact that there is finally enough time between DNA replication in S and chromosome condensation in M.

7. "work from our laboratory showed that H3K27me3 in flies is intergenerationally transmitted from the maternal germline, inherited by the female pronucleus, and maintained throughout the nuclear divisions. While the majority of the H3K27me3 domains only appeared during the minor wave of ZGA, thirty-two H3K27me3-rich domains could be already observed at the early stages of embryonic development and appeared to propagate till ZGA". Again, I'm not sure that the term intergenerational transmission is appropriate, since the oocyte was in meiosis I before fertilization and meiosis II after. From looking at the IF data in Zenk et al 2017, it seems that H3K27me3 is gradually lost pre ZGA and, what is important, is not transmission of H3K27me3 but transmission of Ez. This would be in line with results from Mazo's lab, and also with results in mammals, in which full restoration of H3K27me3 takes many hours after DNA replication. Otherwise, proteins responsible for full restoration of H3K27me3 would have to do so during S and M phases.

8. It may be good for non-drosophilists to mention that Nej is the same as CBP/P300

9. In spite of the fact that most people do this, it may be more appropriate to refer to covalent histone modifications as such, rather than "marks".

10. In the section on the "epigenetic inheritance" of H4K16ac, it may be interesting to mention whether the lack of MOF in oocytes has effects beyond not being able to pass this modification to the zygote. Is transcription normal during oogenesis and are all RNAs normally present in mature oocytes the same as those in oocytes lacking MOF? Also, is it really known that H4K16ac "prime(s) the activation.....by eliciting nucleosome accessibility" or is this just assumed?

11. "the genome is not ready to be pervasively expressed and even though some loci start showing active epigenetic marks, these will only be instructive at later stages". Accumulating evidence, including from the authors lab (Ciabrelli et al Sci Adv 2023), questions the instructive role of active histone modifications in transcription. See also PMID: 35668298 for another example.

12. "The subnuclear distribution of active and inactive regions establishes the so-called "A" and "B" compartments, respectively". Authors should describe a more recent view of compartments. Obviously, regions with H3K27me3 do not interact with regions containing H3K9me3 or active regions, so there should be at least 3 compartments, no matter what Eigenvectors appear to say. Also, as the authors mention, Drosophila TADs contain co-regulated genes, either active or inactive. If there is little transcription, before ZGA, TADs are not visible in Hi-C data because each chromosome is one large TAD of inactive genes.

13. Although it may be correct to say that H3K9me2/3 is a hallmark of constitutive heterochromatin, as it is often stated, it is not true that all regions with H3K9me2/3 are constitutive heterochromatin. In Drosophila these histone modifications are associated with active transcription, as well as HP1. In mammals, enhancers and promoters containing discrete H3K9me3 sites are reprogrammed during cell differentiation and they get demethylated to H3K9me2, the DNA at the site becomes demethylated, and adjacent genes become active. This "heterochromatin" is not, therefore, constitutive.

14. "Polycomb-mediated repression spreads on chromatin via deposition of the H3K27me3 mark during the minor wave of ZGA". Is there evidence for actual "spreading" in this case?

15. "Histone H1 is an evolutionary conserved chromatin protein, which binds both to intranucleosomal DNA and to linker DNA and contributes to the formation of 30 nm higher-order chromatin structures" I may be wrong on this but I was under the impression that experiments using super-resolution microscopy had ruled out the existence of the 30 nm fiber, at least in mammals (see for example PMID: 25768910). Is this not the case in *Drosophila*? Please note that Prendergast and Reinberg 2021 never mentions the 30 nm fiber.
16. "*Drosophila* embryos are loaded with BigH1, whose spreading on chromatin prevents premature transcription". Is this protein already on chromatin in the zygote or is it loaded as the nuclei divide? Is there evidence that H1 actually spreads? What do the authors mean by "BigH1 is distributed on the whole epigenome"? Only on nucleosomes containing histone modifications?
17. The paragraph starting "While Nej- and Gcn5-dependent acetylation marks start to appear during the minor wave of ZGA" suggests that H4K16ac deposited by MSL on the male X, which depends on transcription of roX, does not take place until ZGA, whereas K4K16ac at genes that eventually would be constitutively expressed in autosomes were present in the oocyte and are maintained during the different replication cycles until ZGA. This would require that, at each cycle, half of H4 is acetylated in K16 de novo. At least in mammals, for some active modifications, this requires transcription (PMID: 37468626. Also, <https://doi.org/10.1101/2023.04.19.537523>), and histone acetylation is a consequence, not a cause, of RNAPII, which is responsible for recruiting and promoting the activity of histone acetyltransferases (PMID PMID: 33431884). Is H4K16ac an exception?
18. When ATAC-seq is performed in nuclear cycles 11-13 embryos, do the results represent what is happening in replicating chromatin or in mitotic chromosomes?
19. "it's binding to chromatin is temporarily lost during mitosis" should be "its"
20. "At nuclear cycle 12, TAD borders are defined by Nej-dependent acetylation marks (i.e. H3K18ac, H3K27ac, and H4K8ac), while H3K9ac and "active" methylation marks (i.e. H3K4me3, H3K36me3) will appear on TAD boundaries only at nuclear cycle 14 (Hug, Grimaldi et al. 2017)". Does the term "are defined" imply causality? Would "correlate" be a better explanation of the observations?
21. "interactions between homologous chromosomes spread genome-wide". Does this mean that pairing starts at one site in a chromosome and then moves along the chromosome in a stepwise manner or does it start simultaneously at all Zld sites? Pairing between homologous chromosomes has been mentioned several times and it would be nice to described the origin, significance and consequences of pairing in more a mechanistic and less descriptive manner.
22. "At pericentromeric regions of the current genome version (dm6), H3K9me3 is fully established at major ZGA". Does this mean that centromeres of mitotic chromosomes lack H3K9me3 during the first 12 nuclear cycles? What is known about replication timing in *Drosophila* embryos and are active and inactive regions replicated at the same time, which may affect the ability to visualize H3K9me3?
23. "pointing out that H3K9me2/3 presence at early stages is required for embryonic development". As alternative explanations, could it be that H3K9me3 is required in the oocyte to properly express RNAs or proteins required for embryonic development, or for proper meiosis independent of what happens in the embryo?
24. "but they do not spread along the chromosome arms". Does HP1 bind to different sites in chromosomes but without spreading?
25. "constitutive heterochromatin at pericentromeric regions and contributes to the spatial compartmentalization of inactive regions (B-compartment)". This sentence implies that all inactive regions, including those containing H3K27me3, interact with the pericentromeric heterochromatin. Is this correct?
26. The sentence "whereas moving these H3K27me3 blocks posteriorly are progressively lost in a stepwise fashion" is difficult to understand, something must be missing.
27. "the pervasive silencing of the embryonic genome". This makes it sound as if the genome of the embryo is actively silenced. This may not be the case.
28. "these studies seem to suggest that the high molecular weight of this transcriptional coactivator, which is expressed at relatively high levels (Bonnet, Lindeboom et al. 2019), rather than its catalytic activity, could be crucial for gene regulation and embryonic development". High molecular weight meaning that the protein has several domains that can interact with other proteins? Or are the high levels what is important? Does Nej bind RNA, like mammalian P300?
29. The statement "Although most of the genome seems to be occupied by H3K23ac in cell culture conditions (Feller, Forne, et al. 2015), only a few genes are affected by its absence during ZGA in living embryos". This statement that the reason for the

differences is cell culture versus normal embryo. Alternatively, it could be that the cells used for tissue culture represent a small population of all the cells in the embryo and their contribution cannot be observed in bulk RNA-seq studies of whole embryos.

30. The sentence "protamine-to-histone whole epigenome replacement" implies that protamines carry epigenetic information, since it seems to imply that there is an epigenome when there are mostly protamines.

31. In the section on "Chromatin accessibility", authors should consider that ATAC-seq measures transcription factor occupancy. During *Drosophila* oogenesis, many genes are regulated at the level of promoter-proximal pausing by controlling the release of RNAPII. Therefore, for many genes, the transcription complex is already present at the promoter, and ATAC-seq signal is not expected to change between transcriptionally active and inactive, as measured by RNA levels from RNA-seq. This issue is further affected by the fact that a large proportion of *Drosophila* genes are less than 500 bp from adjacent genes and in the opposite orientation, which makes it difficult to distinguish which promoter corresponds to the observed ATAC-seq signal. Please also note that differences in ATAC-seq signal may not indicate "degree of openness" but rather the fraction of cells containing a bound transcription factor at the site. Given what is known about the role of RNAPII release in controlling gene expression during *Drosophila* embryogenesis, finding 1/4 of differential ATAC-seq sites between different regions of the embryo may be what is expected. Authors mention this issue later in the review but it should be taken into consideration when discussing changes in ATAC-seq.

32. "some studies showed how the presence of transcription factor binding sites alone could not completely explain these differences in chromatin accessibility between early embryo nuclei (Haines and Eisen 2018) (Hannon, Blythe et al. 2017)" I didn't go back to these papers but the conclusion does not make sense. When looking at the ends of subnucleosomal reads from ATAC-seq experiments, all reads have one end at the TF binding site and a second end next to the adjacent nucleosome. There is no such a thing as accessible chromatin without a bound transcription factor in ATAC-seq experiments because reads corresponding to Tn5 insertions in linker regions lacking bound transcription factors are too small and are lost during bead cleanup of the libraries.

33. In the paragraph starting with "Chromatin accessibility at this stage is also achieved by the action of pioneer factors", authors mention that pioneer factors are necessary to expose enhancers to allow binding of transcription factors but in subsequent text they only mention the pioneer factors. Are any of these other transcription factors known?

34. It should be mentioned that H2Av is the same as H2A.Z in mammals.

35. "The temporal correlation between increased RNA Pol II activity and TAD establishment does not bear a causal link. Indeed, chemical inhibition of the RNA Pol II activity does not interfere with TAD formation. Thus, TAD establishment is independent of transcriptional onset". TAD formation does not depend on RNAPII activity but it does depend on the presence of RNAPII. Chemical inhibition, if it does not result in the depletion of RNAPII occupancy, is not expected to affect TADs, since TADs observed by Hi-C are a representation of interactions among proteins. The finding that "The temporal correlation between increased RNA Pol II activity and TAD establishment does not bear a causal link. Indeed, chemical inhibition of the RNA Pol II activity does not interfere with TAD formation. Thus, TAD establishment is independent of transcriptional onset" can be explained because some RNAPII inhibitors cause the degradation of this protein.

36. Please introduce the concept of tethering elements, which is not a household term and nobody but Mike Levine has ever seen. Same for PREs.

37. "TAD boundaries prevent spurious interactions between enhancers and silencers harbored in distinct TADs, thus compartmentalizing de facto the genome in functional units". At least in mammals, when one looks carefully, one can observe interactions between sequences in different TADs.

38. "Moreover, H3K9me2 regions embedded in chromosome arms, including transposable elements, show preferential interactions with the apical chromocenter, where pericentromeric heterochromatin resides". Do these regions interact because they also have H3K9me3 and/or HP1? It is important to clarify this issue because in mammals, regions containing H3K9me2 alone, no H3K9me3 or HP1, do not interact.

39. "HP1a depletion reduces inter-chromosomal contacts while increasing intra-chromosomal contacts and impairing the proper segregation of the A and B compartments". When comparing Hi-C matrices between two conditions, the data is normalized to equal number of valid contacts. If inter-chromosomal contacts increase, intra-chromosomal contacts have no choice but to decrease. When the authors mention that lack of HP1 only affects a handful of genes and repeats, are the repeats present at thousands of copies? This could explain why there is a big change in 3D organization but a small change in transcription. Also, do sequences in the B compartment become transcribed when HP1 is absent or are they still repressed? Please elaborate more on why "HP1a's role at this particular developmental stage is mostly structurally different from differentiated somatic cells".

40. I would suggest that authors refrain from ending each paragraph by saying that more work is needed to understand what was described in the corresponding paragraph, since this is true for anything in biology or any other discipline.

Referee #1:

Ciabrelli et al have reviewed the role of chromatin regulators and histone modifications in early embryonic development of *Drosophila*. It is a nice summary, but would benefit from a discussion about outstanding questions. For example, how histone marks are transmitted during the first rapid nuclear divisions, and why some marks appear to be intergenerationally inherited and others not.

We thank Referee #1 for the time spent reviewing our manuscript and for his/her suggestions. We have now added discussions on the points raised, and we have addressed all the reviewer's comments.

Comments:

1. The authors have structured the review into 3 parts; Fertilization and early divisions, minor ZGA and major ZGA. Although I understand the thought behind this, to me it feels repetitive to discuss heterochromatin, chromatin architecture etc multiple times. The authors may consider a short outline of the timeline of events, followed by a discussion of each topic (chromatin mark inheritance, constitutive heterochromatin etc).

We thank Referee #1 for this suggestion. Indeed, we had considered this alternative while writing the manuscript. However, we intentionally chose to organize the review by emphasizing the developmental progression, including stages such as fertilization, early divisions, minor ZGA, and major ZGA. This approach allowed us to discuss relevant events within each developmental stage in their respective chapters. Had we structured the main chapters around molecular topics, such as transcriptional activation, transcriptional silencing, and chromatin architecture, we would have lost the developmental perspective we wanted to convey to our readers. Additionally, this would have led to repetitive introductions of each developmental stage in every chapter that discussed these chromatin topics.

2. In the Chromatin mark inheritance section, the recent paper from the Loppin lab describing a role for the paternal loss protein in evicting H3-H4 tetramers from spermatid DNA should be mentioned (PMID: 37943933).

We have added the suggested reference.

3. I could not understand why CID and Homothorax is discussed in the context of constitutive heterochromatin (p. 5, it is annoying that page and line numbers are missing from the ms).

The reason why CID and Homothorax are discussed in the context of constitutive heterochromatin is that their presence is necessary for the proper assembly of the centric/centromeric heterochromatin during preblastodermic divisions, as shown in PMID: 19652544. We have however rephrased the paragraph to make it more clear.

4. Further down on same page, the section that starts with Histone replacement...pointed out that E(z) enzymatic activity.... should be replaced with E(z) histone methyltransferase activity....

We have modified the text as suggested.

5. In this section Nejire is mentioned for the first time. For non-Drosophila readers, it would be beneficial to point out that this is the homolog of p300/CBP.

We have modified the text as suggested.

6. At the bottom of this page and top of next page, it is suggested that H3K27me3 is necessary to protect enhancers from Nej binding. However, this model is in conflict with evidence showing that Nej occupies many H3K27me3 regions (PMID: 36206738).

Thank you for pointing out this discrepancy. In our manuscript, we suggested that H3K27me3 is necessary to protect enhancers from Nej binding. We recognize the evidence indicating that Nej occupies many H3K27me3 regions (PMID: 36206738) and appreciate the opportunity to clarify our statement. Upon re-evaluation and consideration of the reviewer's comment, we suggest that while H3K27me3 may play a role in modulating Nej binding, it is not an absolute barrier. This interpretation aligns with the existing evidence and accommodates the complexity of the interaction between Nej and H3K27me3. We have revised our text to reflect this nuanced understanding and to avoid any potentially conflicting wording. We hope this clarification addresses the reviewer's concerns and provides a more accurate depiction of the current scientific consensus.

7. Top of p. 6 should also include the following reference (PMID: 25669886) when discussing the centrality of H3K27me3 in gene silencing.

We have added the suggested reference.

8. The authors claim that histone modifications play instructive roles (e.g. H4K16ac on p. 6). What do they mean by this? Clearly, histone modification cannot direct cell-specific gene expression without transcription factors.

Deposition of H4K16ac mark was shown to be necessary for gene activation in flies in certain contexts (Copur, Gorchakov et al. 2018). This is in contrast with the function of other acetylation marks, whose functionality has been tested through histone replacement systems or by using catalytic inactive enzymes. As Referee #1 correctly points out, H4K16ac presence is not sufficient for gene activation if not supported by the action of TFs. In other words, its presence is necessary but not sufficient in certain contexts, unlike other

acetylation marks whose presence appears to be not necessary for gene expression. In order to avoid ambiguity, we removed the word “instructive” from the text.

9. It would be interesting to read the authors' view on why H3K9me3 and H3K27me3 increase dramatically at the onset of ZGA.

We have added this point to the discussion.

10. I'm missing a discussion about how histone marks that are deposited on the maternal chromosome could be propagated during the nuclear divisions that follow fertilization.

We have added this point to the discussion.

11. The description of HDAC3 on p. 14 is a little misleading. HDAC3 does not cooperate with CtBP. Instead, there are 2 co-repressor complexes involved in Snail-mediated repression, CtBP that interacts with HDAC1, and Ebi/SMRTER that interact with HDAC3. The authors may also want to mention that HDAC3 has both catalytic and non-catalytic roles in early embryo development (PMID: 37455638).

We thank Referee #1 to have raised this point. By cooperation we did not mean a direct interplay between HDAC3 and CtBP in Snail-mediated repression, but rather two parallel mechanisms that work in the same direction. In order to not mislead readers, we have rephrased the paragraph. We have also mentioned the recent paper on HDAC3 (PMID: 37455638)

12. It is strange to mention the co-repressor Grunge without mentioning Brakeless. Furthermore, a really key co-repressor in the early embryo is Groucho that should also be included.

We thank Referee #1 for this suggestion. We added Groucho and Brakeless.

13. This is a minor point, but the twisted phenotype resulting from nejire mutations is not due to downregulation of twist as originally proposed, but to defects in Dpp-signaling (see e.g. PMID: 14550792).

We thank Referee #1 for this clarification. We have added this information in the text.

14. When discussing TrxG proteins in gene activation (p. 16 and 17), Brm and other SWI/SNF components are missing, although they are discussed in a later section.

Indeed, we introduce chromatin remodeling in the later section, and we mention that Brm and other SWI/SNF components are also classified as TrxG proteins. Nonetheless, we have added a sentence about chromatin remodeling when we first introduce TrxG proteins in the text

15. Differences in chromatin accessibility along the DV axis were recently described (PMID: 38166964), and could be contrasted to the data from the AP axis mentioned in the manuscript.

We have added this recent article to the manuscript as suggested.

Referee #2:

Overall, this is a nice, extensively cited review focusing on how chromatin structure is reorganized during the maternal-to-zygotic transition in *Drosophila*. Despite numerous recent reviews on the MZT and this transition in *Drosophila* more specifically, the focus of this review on chromatin is distinctive and is an area that has been relatively less well documented than gene expression dynamics. As such, this review will be an important resource for many researchers.

One major concern with the review is the numerous discussions and 16 citations to Atinayeva et al. This appears to be an upcoming manuscript from the Iovino lab but is not available online (either published at a peer reviewed journal or on bioRxiv). Thus, it is impossible to assess the data in this manuscript and how it impacts the current review. As such, it was not possible to review this aspect of the review. Similarly, the authors cite unpublished data from the lab. Again, this lack of transparency limits the ability of the reviewer to assess the data and therefore the conclusions that can be drawn. All the data discussed in a review should be available to the reviewer. That was not the case for this review.

We thank Referee #2 for appreciating the novelty and completeness of our review. In response to their comments, we have now added the citation for Atinbayeva et al. We removed all mentions of unpublished data from our lab within the manuscript. We have also refined the text based on the specific feedback from Referee #2.

Atinbayeva's work is currently in press with the EMBO Journal, and we had hoped it would be available online before the review process. Unfortunately, there have been delays, and it is not yet online. Therefore, we have attached the manuscript to this current revision for reference.

Specific comments:

1) Figures were never referenced in the body of the manuscript. As such, it was challenging to determine how they were increased the clarity of the review.

We have now added references for the figures inside the body of the manuscript.

2) It is confusing that the Introduction only covers a discussion of zygotic genome activation, despite the fact that heterochromatin establishment is extensively covered. The authors should consider mentioning the heterochromatin in the Introduction.

We have now mentioned heterochromatin and chromatin structure in the introduction.

3) The mention that "the maternally provided pool of mRNAs inevitably declines" in Major ZGA; chromatin-mediated transcriptional silencing makes this sound like a passive process. It would be important to rephrase so that it is clear that ZGA is actively tied to this degradation.

We have modified the text as suggested.

4) In the discussion of mitotic retention, it could be useful to discuss the challenges posed by mitosis and the fact that early expression of the pioneer factor Zelda can't activate early gene expression (Larson et al. G3 2022).

We have included this information in the text.

5) Nej is first defined in Fertilization and early divisions; chromatin-mediated transcriptional silencing. But the reference is to the naming being related to twisted in Japanese is not until later. Perhaps reference this during the initial mention of the protein. In addition, a few lines below the reference to twisted "gene" is italicized after the gene twist when it should not be.

We have modified the text as suggested at both points.

6) The Discussion rather than summarizing and bringing together the data brings up the entirely new concept of transgenerational inheritance of chromatin marks. This is an important area that the Lovino lab has studied extensively. Perhaps this would be best to include as a stand-alone section prior to a Discussion.

We thank Referee #2 for this suggestion. We have now done as the reviewer suggests.

Some of the text in the figures is too small to be legible.

We have fixed it.

7) In Figure 1: H3K27me3 and H3K9me3 levels appear close to maximum at stage 2. This is not what the data show. There is a clear increase in these marks at stage 5. This should be reflected in the figure. The authors may also want to indicate H1 levels.

We have improved Figure 1 as suggested and also split Figure 1 into three figures.

8) In Figure 2: it could be good to label the male and female flies for a non-Drosophila audience.

We have modified the figure as suggested.

9) While the authors should be commended for their well-cited manuscript, there are a selection of places that additional citations should be added:

Introduction:

Lott et al. PLoS Biol 2011 should be cited in the Introduction regarding the minor wave of zygotic genome activation.

Heyn et al Cell Rep 2014 should be cited for genes being short and lacking introns.

Soluri et al. eLife 2020 should be cited with Koromilla et al. for the role of Odd-paired.

We have added the suggested references in the manuscript.

10) Fertilization and Early Divisions; Chromatin-mediated transcriptional activation

Ali-Murthy et al. PLoS Genet 2013 should be cited regarding earliest observed transcription.

ten Bosch et al Development 2006 would be a useful citation regarding the timing of specific patterning and sex-determination genes.

Li et al. eLife 2014 should be cited along with Chen and Samata for histone modifications.

We have added the suggested references in the manuscript.

11) Major ZGA; Constitutive heterochromatin

Seller et al. Genes Dev 2019 should be cited for the role of histone methyltransferases in heterochromatin formation.

We have added the suggested reference in the manuscript.

12) Major ZGA; chromatin-mediated transcriptional activation

Quite late in this section is a reference to Li, Harrison et al 2014. This seems to be incorrectly placed. It should be deleted and the appropriate reference included.
Hunt et al. Mol Cell 2022 should be mentioned to support the role of catalytic independent roles of Nej and the balance with H3K27me3.

We have added the suggested reference in the manuscript.

13) Major ZGA: chromatin accessibility

It should be mentioned that promoters are generally accessible regardless of transcriptional state.

Liang et al. Nature 2008 should be included with Ibarra-Morales, Rauer et al 2021 for the ability of Zelda to broadly activate gene expression.

We have added the suggested reference in the manuscript and included the suggested sentence.

14) "Among them, the ATPase subunit of the SWI/SNF complex Brahma, .. ." is a sentence fragment.

We have rephrased the text.

15) There is a completely italicized sentence that should not be italicized in this section.

We have modified the text as suggested.

16) Major ZGA: chromatin architecture

It could be useful to initially explain tethering elements as they are relatively newly defined chromatin structures.

We have now introduced what is defined as a tethering element inside the manuscript.

17) The role of GAF in establishing heterochromatin during this time should be discussed in relation to H3K9me3 and HP1a. Li et al. Mol Cell 2023 and Gaskill et al. Dev Cell 2023 would be good to discuss.

We did discuss the role of GAF in heterochromatin establishment during Major ZGA:
"Also, the insulator protein and pioneer factor GAF exerts an important role in heterochromatin formation at specific loci during early embryogenesis. This protein binds to AAGAG satellite repeats, contributing to their silencing and heterochromatin establishment (Gaskill, Soluri et al. 2023)"

Moreover, we mention Li et al. Mol Cell 2023 in the "chromatin architecture" section

Referee #3:

The manuscript is an encyclopedic review of the fields of chromatin and transcription during early *Drosophila* embryogenesis. The authors cover all possible topics with an emphasis on the more recent literature, which is cited in a balanced manner. I think the review will be helpful and a reference for those interested in this topic. I went through the manuscript with a microscope and I have a lot of suggestions that the authors may think are too picky. Many of the suggestions, which correspond to questions elicited by my reading, may be a consequence of how the manuscript is written. I think the goal of the authors is to have everything known about the topic in one place. As a consequence, the manuscript reads as a collection of many facts linked to each other by the topic under discussion in each section. From my perspective, and I recognize that this may be my own biased view, the authors do not attempt to distil the information of what is known and attempt to gain a mechanistic understanding of what is happening during *Drosophila* embryogenesis. The understanding of the relationship between chromatin and transcription is much more detailed and sophisticated in mammals. The *Drosophila* field would benefit by attempting to explain what happens in *Drosophila* in the context of what is known in mammals. A second issue is that the authors take what has been published at face value, without critically evaluating whether the interpretation of things that were published 5 years ago is still correct, and without attempting to re-interpret older results in the context of what we know today in order to draw a more mechanistic understanding of the topic. The following is a list of things that popped in my mind while reading the manuscript:

We thank Referee #3 for appreciating and thoroughly examining our manuscript. Replying to Referee #3's comments has helped us to improve it significantly.

1. "Its genome is largely dormant". Transcriptionally dormant may be a more appropriate term. The genome is replicating once every 8-10 min during this time, so it's busy doing other things.

We have modified the text as suggested.

2. Since the review is probably written for a large audience, not just *Drosophila* specialists, it may be better to avoid using specialized terms, like Bownes' stages. It would more intuitive to the general reader to refer to stages in terms of nuclear cycles.

We agree with Referee #3 regarding the technicality of introducing Bownes' stages. However, many studies cited in subsequent sections were conducted at specific developmental stages rather than during particular cell cycles. We believed that defining Bownes' stages in the introduction would provide readers with a clearer understanding of the developmental windows we frequently reference throughout the manuscript. Additionally, Figure 1 illustrates a correspondence between these stages and nuclear cycles. We have now included this information in the text, allowing readers unfamiliar with *Drosophila* research to refer to the figure for guidance on early developmental stages.

3. The description of the replacement of protamines for histones in the male pronucleus is described as if the embryo specifically wants H3.3 to be present in the paternal but not the maternal chromosome. This histone H3.3 does not appear to play any specific role in embryonic development based on the description, and it is eventually replaced by H3. I wonder if this is just a consequence of what is available in the zygote at the time. Are histone H3.3, ASF1, and HIRA the only ones present in the oocyte or are canonical histone proteins and their chaperones also present? The oocyte must have a lot of mRNAs encoding canonical histones. Do these RNAs have to be translated to provide histone proteins for the next S-phase or are the proteins already present? I imagine the oocyte nucleus is going through meiosis II, so it is not surprising that it keeps the histones it already had.

H3 vs H3.3 asymmetry in maternal versus paternal chromatin reflects the replication dependent vs replication-independent modality of histone incorporation. Both H3 and H3.3 with their respective chaperones are present in the early zygote. However, protamine replacement does not occur during S phase, when canonical H3 is incorporated in newly assembled nucleosomes. Therefore, the difference is the cell-cycle timing of histone incorporation.

4. There is no reference in the list for work cited as Atinbayeva et al. When the authors state that "we demonstrated that H3K9me3 is not de novo established, rather it is intergenerationally inherited from the oocyte", is this another way of saying that the oocyte was in meiosis I, it was then fertilized, it now goes through meiosis II, and it kept the H3K9me3 it had before? In the discussion of heterochromatin and H3K9me2/3, authors should keep in mind work for the Bas van Steensel's lab showing that these covalent modifications are present at actively transcribed genes in what is normally considered to be euchromatin.

We have now added the missing reference from Atinbayeva et al. to the list and attached the manuscript, which is currently in press.

Regarding the distribution of H3K9me3, the maternal oocyte maintains this modification across meiosis I and II. However, we have no evidence of H3K9me3 presence in euchromatin regions in early embryos. At stage 17, there is some accumulation of H3K9me3 at the transcription start sites (TSS) of genes, but this is not observed in early embryos, where this modification is exclusively found in repetitive regions.

5. It would be interesting to consider what is happening to heterochromatin in the early embryo in more detail. Does the formation of heterochromatin depend on the formation of biomolecular condensates? If so, what does it mean that HP1 is absent from regions containing H3K9me3? The authors consider H3K9me2 and H3K9me3 as being equivalent, but regions containing H3K9me2 do not interact with regions of H3K9me3, suggesting that they may not participate in the same biomolecular condensates. Do the two modifications overlap in ChIP-seq results using antibodies specific to each one?

We thank Referee #3 for her/his intriguing questions. The dependence of HP1 biocondensate and heterochromatin formation needs further investigation. Our Cut@TAG

data with H3K9me2 and H3K9me3 antibodies show a large overlap between these two marks in early *Drosophila* embryos. The reviewer will be able to appreciate this in the attached manuscript from Atinbayeva et al., which is currently in press at *Embo Journal*.

6. The review describes the standard view of transcription in the *Drosophila* embryo i.e., that something special takes place at the ZGA and pioneer factors decide it's time to act. However, an equally plausible scenario is that everything is in place to start transcription in the zygote (see, for example, PMID: 23593026) but transcription does not happen because the short cell cycles do not allow for a sufficiently long interphase to complete transcription. In this view, there is nothing special about ZGA other than the fact that there is finally enough time between DNA replication in S and chromosome condensation in M.

We agree with Referee #3 on the importance of interphase lengthening. Data indicate that the rapid nuclear cycles observed in the earliest divisions are only compatible with the expression of a few short, intron-less genes, as discussed in our review. However, the activity of pioneer factors and H2Av deposition are crucial for ZGA waves, as shown by studies on mutants. In summary, while the lengthening of nuclear cycles alone is not sufficient to activate thousands of genes during ZGA, it is necessary. Additionally, remodeling activity by ATPase enzymes, though not fully elucidated, and issues with accessibility can also hinder transcription factor binding in early embryos.

7. "work from our laboratory showed that H3K27me3 in flies is intergenerationally transmitted from the maternal germline, inherited by the female pronucleus, and maintained throughout the nuclear divisions. While the majority of the H3K27me3 domains only appeared during the minor wave of ZGA, thirty-two H3K27me3-rich domains could be already observed at the early stages of embryonic development and appeared to propagate till ZGA". Again, I'm not sure that the term intergenerational transmission is appropriate, since the oocyte was in meiosis I before fertilization and meiosis II after. From looking at the IF data in Zenk et al 2017, it seems that H3K27me3 is gradually lost pre ZGA and, what is important, is not transmission of H3K27me3 but transmission of Ez. This would be in line with results from Mazo's lab, and also with results in mammals, in which full restoration of H3K27me3 takes many hours after DNA replication. Otherwise, proteins responsible for full restoration of H3K27me3 would have to do so during S and M phases.

We believe that the term "intergenerational" transmission is appropriate, as it specifically refers to non-DNA inheritance from one generation to the next. This is distinct from "transgenerational" transmission or transgenerational epigenetic inheritance, which involves the inheritance of epigenetic information across multiple generations (Fitz-James and Cavalli 2022).

The transmission of E(z) or PRC2 alone would not be sufficient without the pre-existing load of H3K27me3 on the chromatin. This insufficiency might also be due to timing issues, as an embryo without pre-formed domains would not have enough time to establish them from scratch immediately after fertilization. Furthermore, studies on E(z) mutants by Zenk et al. demonstrate that within about two divisions, all inherited H3K27me3 signals become

undetected, strongly supporting the necessity of active enzymatic activity early on to maintain these inherited domains.

While the inheritance of H3K27me3 and H3K9me3 plays a crucial developmental role in regulating early embryogenesis, we speculate that environmentally induced chromatin changes in the germline could exploit the same developmental mechanisms. Such changes might transmit environmental information along with developmental information to the next generation, thus serving as an epigenetic template for conveying environmental influences across generations.

8. It may be good for non-drosophilists to mention that Nej is the same as CBP/P300

We have added this information to the manuscript.

9. In spite of the fact that most people do this, it may be more appropriate to refer to covalent histone modifications as such, rather than "marks".

We thank Referee#3 for this suggestion; we have changed it.

10. In the section on the "epigenetic inheritance" of H4K16ac, it may be interesting to mention whether the lack of MOF in oocytes has effects beyond not being able to pass this modification to the zygote. Is transcription normal during oogenesis and are all RNAs normally present in mature oocytes the same as those in oocytes lacking MOF? Also, is it really known that H4K16ac "prime(s) the activation.....by eliciting nucleosome accessibility" or is this just assumed?

The mature oocyte is transcriptionally silent, thus the absence of MOF does not affect its transcriptional state. However, it is unclear whether MOF deficiency impacts the transcriptional state of nurse cells, which are responsible for the maternal mRNA load in the embryo.

To determine whether transcriptional changes in nurse cells are responsible for the phenotype observed in early embryos, RNA sequencing experiments should be conducted on unfertilized eggs, similar to our previous studies in Zenk et al., Ciabrelli et al., and Atinbayeva et al. This would help clarify whether the phenotype results from chromatin changes that affect ZGA and de novo transcription, rather than changes in nurse cell transcription.

Additionally, experiments using H4K16R transgenic flies have replicated the phenotypes observed with MOF depletion (Samata, Alexiadis, et al. 2020).

ATAC-seq experiments conducted before ZGA indicate that the lack of maternal MOF, and consequently the absence of the maternally maintained H4K16ac mark, leads to a global decrease in nucleosome accessibility in stage 3 to stage 4 embryos. Similarly, the depletion

of maternal Msl-3 mimics the effects of MOF depletion. The majority of MOF-sensitive ATAC-seq peaks were found within H4K16ac-positive genes (Samata, Alexiadis, et al. 2020).

11. "the genome is not ready to be pervasively expressed and even though some loci start showing active epigenetic marks, these will only be instructive at later stages". Accumulating evidence, including from the authors lab (Ciabrelli et al Sci Adv 2023), questions the instructive role of active histone modifications in transcription. See also PMID: 35668298 for another example.

We agree with Referee #3 that most active modifications are not instructive, including the ones studied from our lab. Nonetheless data show that H4K16ac represents an exception to this rule. We do agree that the term "instructive" might be misleading as it could infer that H4K16ac presence is sufficient to trigger transcription, which clearly is not. As also suggested by Referee #1, we removed the word "instructive" from the text to avoid any confusion

12. "The subnuclear distribution of active and inactive regions establishes the so-called "A" and "B" compartments, respectively". Authors should describe a more recent view of compartments. Obviously, regions with H3K27me3 do not interact with regions containing H3K9me3 or active regions, so there should be at least 3 compartments, no matter what Eigenvectors appear to say. Also, as the authors mention, *Drosophila* TADs contain co-regulated genes, either active or inactive. If there is little transcription, before ZGA, TADs are not visible in Hi-C data because each chromosome is one large TAD of inactive genes.

We thank Referee#3 for this comment. We have included recent studies in mammalian cells, which subdivided A and B compartments into sub-compartments. Specifically, we included work in human cells that divided B-compartment into three types, therefore differentiating H3K9 methylation and H3K27 methylation-marked regions.

The information about TADs containing co-regulated genes is general information that does not imply the presence of TADs before ZGA. To make it clear, we changed the sentence in line... "In *Drosophila*, TADs were first identified in major ZGA embryos".

13. Although it may be correct to say that H3K9me2/3 is a hallmark of constitutive heterochromatin, as it is often stated, it is not true that all regions with H3K9me2/3 are constitutive heterochromatin. In *Drosophila* these histone modifications are associated with active transcription, as well as HP1. In mammals, enhancers and promoters containing discrete H3K9me3 sites are reprogrammed during cell differentiation and they get demethylated to H3K9me2, the DNA at the site becomes demethylated, and adjacent genes become active. This "heterochromatin" is not, therefore, constitutive.

We agree with Referee #3 that the presence of H3K9me2/3 is not sufficient to determine what heterochromatin is and is not. A clear example is *Drosophila* chromosome 4, which is completely covered by H3K9me2/3. We modified the text in order not to create this misunderstanding

14. "Polycomb-mediated repression spreads on chromatin via deposition of the H3K27me3 mark during the minor wave of ZGA". Is there evidence for actual "spreading" in this case?

With the term spreading, we refer to the widening of H3K27me3 domain that could be observed by ChIP experiment or Cut&TAG experiments between earlier and later embryonic stages

15. "Histone H1 is an evolutionary conserved chromatin protein, which binds both to intranucleosomal DNA and to linker DNA and contributes to the formation of 30 nm higher-order chromatin structures" I may be wrong on this but I was under the impression that experiments using super-resolution microscopy had ruled out the existence of the 30 nm fiber, at least in mammals (see for example PMID: 25768910). Is this not the case in Drosophila? Please note that Prendergast and Reinberg 2021 never mentions the 30 nm fiber.

We have removed any mention of 30 nm fiber formation. Our claim referred to older in vitro studies, but it has never been tested in flies.

16. "Drosophila embryos are loaded with BigH1, whose spreading on chromatin prevents premature transcription". Is this protein already on chromatin in the zygote or is it loaded as the nuclei divide? Is there evidence that H1 actually spreads? What do the authors mean by "BigH1 is distributed on the whole epigenome"? Only on nucleosomes containing histone modifications?

We are not aware about data showing BigH1 presence on chromatin at the zygote stage or lack thereof. We agree with Referee #3 that it would be quite interesting to know. By "BigH1 is distributed on the whole epigenome" we actually meant the whole genome. We have now corrected this word in the text.

17. The paragraph starting "While Nej- and Gcn5-dependent acetylation marks start to appear during the minor wave of ZGA" suggests that H4K16ac deposited by MSL on the male X, which depends on transcription of roX, does not take place until ZGA, whereas K4K16ac at genes that eventually would be constitutively expressed in autosomes were present in the oocyte and are maintained during the different replication cycles until ZGA. This would require that, at each cycle, half of H4 is acetylated in K16 de novo. At least in mammals, for some active modifications, this requires transcription (PMID: 37468626). Also, <https://doi.org/10.1101/2023.04.19.537523>), and histone acetylation is a consequence, not a cause, of RNAPII, which is responsible for recruiting and promoting the activity of histone acetyltransferases (PMID PMID: 33431884). Is H4K16ac an exception?

We agree with Referee #3 that the vast majority of "active" chromatin marks are a mere consequence of transcriptional activity. Nonetheless, evidences from Samata, Alexiadis et al. 2020 show that H4K16ac represents an exception. The mark is visible by IF and ChIP in the early stages, even though the underlying genes are not transcribed yet.

18. When ATAC-seq is performed in nuclear cycles 11-13 embryos, do the results represent what is happening in replicating chromatin or in mitotic chromosomes?

According to material and methods from Blythe and Wieschaus 2016, the embryos for ATAC-seq were hand-picked in interphase after microscope visualization. This is also clear from Figure 1A of their manuscript. The timing and duration of the nuclear cycle at these stages are extraordinarily synchronized and reproducible.

19. "it's binding to chromatin is temporarily lost during mitosis" should be "its"

We have corrected the typo as suggested.

20. "At nuclear cycle 12, TAD borders are defined by Nej-dependent acetylation marks (i.e. H3K18ac, H3K27ac, and H4K8ac), while H3K9ac and "active" methylation marks (i.e. H3K4me3, H3K36me3) will appear on TAD boundaries only at nuclear cycle 14 (Hug, Grimaldi et al. 2017)". Does the term "are defined" imply causality? Would "correlate" be a better explanation of the observations?

We have modified the text as suggested.

21. "interactions between homologous chromosomes spread genome-wide". Does this mean that pairing starts at one site in a chromosome and then moves along the chromosome in a stepwise manner or does it start simultaneously at all Zld sites? Pairing between homologous chromosomes has been mentioned several times and it would be nice to describe the origin, significance and consequences of pairing in more a mechanistic and less descriptive manner.

Homologous chromosome pairing starts at multiple sites, as shown in the cited manuscript (Erceg, AlHaj Abed et al. 2019). Zelda binding sites coincide with early pairing sites, which lead to spreading of pairing. Unfortunately, little is known about the molecular significance and consequences of homologous pairing in dipterans. There are no reports showing how disruption of pairing would lead to significant changes in gene expression. It is possible that homologous pairing is important for other aspects of genome biology in flies.

22. "At pericentromeric regions of the current genome version (dm6), H3K9me3 is fully established at major ZGA". Does this mean that centromeres of mitotic chromosomes lack H3K9me3 during the first 12 nuclear cycles? What is known about replication timing in Drosophila embryos and are active and inactive regions replicated at the same time, which may affect the ability to visualize H3K9me3?

According to our IF data, H3K9me3 is present at mitotic chromosomes during the first 12 nuclear cycles.

23. "pointing out that H3K9me2/3 presence at early stages is required for embryonic development". As alternative explanations, could it be that H3K9me3 is required in the oocyte to properly express RNAs or proteins required for embryonic development, or for proper meiosis independent of what happens in the embryo?

We thank Referee #3 for this suggestion. The oocyte in the late stages of development retains H3K9me3 and is transcriptionally inactive. Furthermore, embryos depleted of H3K9me3 successfully reach and initiate ZGA. We also conducted RNA sequencing on unfertilized eggs to determine whether defects in nurse cell transcription could be responsible for the observed phenotype. Our findings revealed no defects that could explain the observed phenotypes.

24. "but they do not spread along the chromosome arms". Does HP1 bind to different sites in chromosomes but without spreading?

That is right. The peak size of HP1a is similar between pre-ZGA and ZGA embryos at chromosome arms, implying there is no spreading of HP1a at ZGA at chromosome arms shown by Zenk&Zhan et al., 2021.

25. "constitutive heterochromatin at pericentromeric regions and contributes to the spatial compartmentalization of inactive regions (B-compartment)". This sentence implies that all inactive regions, including those containing H3K27me3, interact with the pericentromeric heterochromatin. Is this correct?

We agree with Referee #3 that this causes confusion. Therefore, we have removed the B-compartment from the text and added "contributes to the spatial compartmentalization of inactive pericentromeric regions."

26. The sentence "whereas moving these H3K27me3 blocks posteriorly are progressively lost in a stepwise fashion" is difficult to understand, something must be missing.

We agree with Referee #3 that the structure of this sentence lacks clarity, particularly for non drosophilists. We have now modified it.

27. "the pervasive silencing of the embryonic genome". This makes it sound as if the genome of the embryo is actively silenced. This may not be the case.

We agree with Referee #3 that the use of the word "pervasive" might be misleading. We have modified the text.

28. "these studies seem to suggest that the high molecular weight of this transcriptional coactivator, which is expressed at relatively high levels (Bonnet, Lindeboom et al. 2019), rather than its catalytic activity, could be crucial for gene regulation and embryonic development". High molecular weight meaning that the protein has several domains that can interact with other proteins? Or are the high levels what is important? Does Nej bind RNA, like mammalian P300?

By high molecular weight we refer to the actual molecular weight of the Nejire protein of 332 KDa, which is pretty remarkable in flies. Moreover, this protein is also abundantly expressed during early embryogenesis (Bonnet, Lindeboom et al. 2019) and it has been shown to bind the chromatin on nearly all the active regions during ZGA (Ciabrelli, Rabbani et al. 2023). Therefore, we speculate that its massive presence on chromatin could be crucial for its coactivator function, rather than its catalytic activity. We demonstrated that the NTD region, which contains a KIX domain, is by itself a very powerful transactivator *in vitro* (Ciabrelli, Rabbani et al. 2023). This domain is known to be involved in heterodimerization between mammalian CBP and transcription factors (Parker, Ferreri et al. 1996) (Goto, Zor et al. 2002). We do not know whether Nejire binds directly RNAs in flies, however it is necessary for RNA-DNA hybrid (R-loop) formation and nucleosome depletion at Polycomb Response Elements (PREs) (PMID: 36206738)

29. The statement "Although most of the genome seems to be occupied by H3K23ac in cell culture conditions (Feller, Forne, et al. 2015), only a few genes are affected by its absence during ZGA in living embryos". This statement that the reason for the differences is cell culture versus normal embryo. Alternatively, it could be that the cells used for tissue culture represent a small population of all the cells in the embryo and their contribution cannot be observed in bulk RNA-seq studies of whole embryos.

We agree with Referee #3 that the difference between the systems (cell line vs early embryo) could in principle explain the lack of transcriptional effects observed when Enok is depleted during ZGA. However, we have unpublished data showing how H3K23ac is also massively deposited on chromatin during ZGA. As we cannot mention these unpublished results in the review, we have now added Referee #3's interpretation of the current literature inside the text.

30. The sentence "protamine-to-histone whole epigenome replacement" implies that protamines carry epigenetic information, since it seems to imply that there is an epigenome when there are mostly protamines.

It has been shown that protamines are posttranslationally modified in both flies and mammals. In principle, these modifications might carry epigenetic information. Nonetheless, even in the case that protamine modifications do not have any "epigenetic" function, we believe that the replacement from a naïve epigenetic state (protamine) to a potentially editable epigenetic state (histone) could be defined as an "epigenome replacement"

31. In the section on "Chromatin accessibility", authors should consider that ATAC-seq measures transcription factor occupancy. During *Drosophila* oogenesis, many genes are

regulated at the level of promoter-proximal pausing by controlling the release of RNAPII. Therefore, for many genes, the transcription complex is already present at the promoter, and ATAC-seq signal is not expected to change between transcriptionally active and inactive, as measured by RNA levels from RNA-seq. This issue is further affected by the fact that a large proportion of *Drosophila* genes are less than 500 bp from adjacent genes and in the opposite orientation, which makes it difficult to distinguish which promoter corresponds to the observed ATAC-seq signal. Please also note that differences in ATAC-seq signal may not indicate "degree of openness" but rather the fraction of cells containing a bound transcription factor at the site. Given what is known about the role of RNAPII release in controlling gene expression during *Drosophila* embryogenesis, finding 1/4 of differential ATAC-seq sites between different regions of the embryo may be what is expected. Authors mention this issue later in the review but it should be taken into consideration when discussing changes in ATAC-seq.

We agree with Referee #3 that many genes, particularly those active during ZGA, are regulated at the level of transcriptional pausing. Therefore, Pol II occupancy at the transcription start sites (TSS) does not vary significantly between cell types, despite differences in transcriptional output. Differences in chromatin accessibility, particularly in enhancer regions, are more pronounced and align with the findings of most studies cited in our manuscript. We also concur that variations in ATAC-seq signals often reflect the proportion of cells containing a specific transcription factor rather than the degree of chromatin openness. We addressed these points in our revised review when discussing the diverse transcriptional programs that emerge during ZGA. However, future studies using single-cell ATAC-seq will be necessary to clarify these aspects further.

32. "some studies showed how the presence of transcription factor binding sites alone could not completely explain these differences in chromatin accessibility between early embryo nuclei (Haines and Eisen 2018) (Hannon, Blythe et al. 2017)" I didn't go back to these papers but the conclusion does not make sense. When looking at the ends of subnucleosomal reads from ATAC-seq experiments, all reads have one end at the TF binding site and a second end next to the adjacent nucleosome. There is no such a thing as accessible chromatin without a bound transcription factor in ATAC-seq experiments because reads corresponding to Tn5 insertions in linker regions lacking bound transcription factors are too small and are lost during bead cleanup of the libraries.

These studies did not conclude that TF binding is not important to get accessible chromatin sites. They claim instead that the presence of a given TF motifs cannot explain different chromatin accessibility profiles between different stages, even though the TF actually binds a given locus in all the conditions compared. They conclude that different chromatin environment could explain differences in local chromatin accessibility given by TF binding. In the case of Bicoid (Hannon, Blythe et al. 2017), the authors defined concentration-sensitive and concentration-insensitive binding sites, with the latter found in more accessible chromatin and bound at low concentration by Bicoid.

33. In the paragraph starting with "Chromatin accessibility at this stage is also achieved by the action of pioneer factors", authors mention that pioneer factors are necessary to expose enhancers to allow binding of transcription factors but in subsequent text they only mention

the pioneer factors. Are any of these other transcription factors known?

Yes, these transcription factors are known, particularly the ones that cooperate with the pioneer factor Zelda. The reason they are not mentioned is that several transcription factors need pioneer factors to transactivate genes during ZGA. In contrast, only a few pioneer factors are known so far. For example, products of gap genes, which are zinc finger transcription factors, require Zelda for their function.

34. It should be mentioned that H2Av is the same as H2A.Z in mammals.

We mention it in the introduction: *"...the deposition of the histone variant H2Av (the Drosophila ortholog of mammalian H2A.Z and H2A.X) by the histone chaperone Domino ..."*

35. "The temporal correlation between increased RNA Pol II activity and TAD establishment does not bear a causal link. Indeed, chemical inhibition of the RNA Pol II activity does not interfere with TAD formation. Thus, TAD establishment is independent of transcriptional onset". TAD formation does not depend on RNAPII activity but it does depend on the presence of RNAPII. Chemical inhibition, if it does not result in the depletion of RNAPII occupancy, is not expected to affect TADs, since TADs observed by Hi-C are a representation of interactions among proteins. The finding that "The temporal correlation between increased RNA Pol II activity and TAD establishment does not bear a causal link. Indeed, chemical inhibition of the RNA Pol II activity does not interfere with TAD formation. Thus, TAD establishment is independent of transcriptional onset" can be explain because some RNAPII inhibitors cause the degradation of this protein.

We completely agree with Referee #3.

36. Please introduce the concept of tethering elements, which is not a household term and nobody but Mike Levine has ever seen. Same for PREs.

We have introduced a definition of tethering elements. PRE/TRE were defined in an earlier point inside the manuscript as Polycomb/Trithorax regulatory elements

37. "TAD boundaries prevent spurious interactions between enhancers and silencers harbored in distinct TADs, thus compartmentalizing de facto the genome in functional units". At least in mammals, when one looks carefully, one can observe interactions between sequences in different TADs.

There are specific cases of inter-TAD long range interactions also in flies. However, those are exceptions to the rule. In general, TAD boundaries do prevent spurious interactions between different TADs in flies.

38. "Moreover, H3K9me2 regions embedded in chromosome arms, including transposable elements, show preferential interactions with the apical chromocenter, where pericentromeric heterochromatin resides". Do these regions interact because they also have H3K9me3 and/or HP1? It is important to clarify this issue because in mammals, regions containing H3K9me2 alone, no H3K9me3 or HP1, do not interact.

Unlike in mammals, H3K9me2 and H3K9me3 seem to be mostly enriched at the same genomic locations in early fly embryos. However, the cited study was performed on late-staged embryos. We do not have information about the overlapping of H3K9me2 and H3K9me3 for late-staged embryos.

39. "HP1a depletion reduces inter-chromosomal contacts while increasing intra-chromosomal contacts and impairing the proper segregation of the A and B compartments". When comparing Hi-C matrices between two conditions, the data is normalized to equal number of valid contacts. If inter-chromosomal contacts increase, intra-chromosomal contacts have no choice but to decrease. When the authors mention that lack of HP1 only affects a handful of genes and repeats, are the repeats present at thousands of copies? This could explain why there is a big change in 3D organization but a small change in transcription. Also, do sequences in the B compartment become transcribed when HP1 is absent or are they still repressed? Please elaborate more on why "HP1a's role at this particular developmental stage is mostly structurally different from differentiated somatic cells".

We thank Referee #3 for these insights. Referee #3 is correct in noting that we cannot rule out the possibility that an increase in repeat transcripts might significantly alter the 3D chromatin organization upon HP1a loss. However, it's important to highlight that only certain genes and repeat sequences are misexpressed following HP1a depletion. We have expanded on this in the manuscript, discussing the differing roles of HP1a in early embryos compared to differentiated somatic cells.

HP1a depletion is associated with an increase in short-range intrachromosomal interactions and a decrease in long-range interactions within pericentric heterochromatin and across chromosome arms. Additionally, changes in chromatin compaction have been confirmed using an orthogonal method that involves oligo paint coupled with microscopy, as shown in Zenk, Zhan et al.

40. I would suggest that authors refrain from ending each paragraph by saying that more work is needed to understand what was described in the corresponding paragraph, since this is true for anything in biology or any other discipline.

We have now removed these vague statements as suggested. We have kept only specific suggestions on what exactly could be done to improve our knowledge in the field.

Dr. Nicola Iovino
MPI-IE Freiburg
Chromatin regulation
Stübeweg, 51
Freiburg
Germany

Dear Nicola,

I am pleased to inform you that your review has been accepted for publication in EMBO reports. Your manuscript will be processed for publication by EMBO Press. It will be copy edited and you will receive page proofs prior to publication.

You will soon be contacted by Springer Nature to sign your publishing license. When you login to the customer service website, please use the following token to waive the article publication charges: LTE3NTC1NJYZMDG. Should you experience any difficulty, please email publishing@embo.org.

Thank you again for your contribution to EMBO Reports!

Best wishes,
Esther
